# A stochastic epigenetic switch controls the dynamics of T-cell lineage commitment

Kenneth KH Ng[1,2], Mary A Yui[2], Arnav Mehta[2†], Sharmayne Siu[3], Blythe Irwin[1], Shirley Pease[2], Satoshi Hirose[2‡], Michael B Elowitz[2,4*], Ellen V Rothenberg[2*], Hao Yuan Kueh[1,2*]

[1]Department of Bioengineering, University of Washington, Seattle, United States; [2]Division of Biology and Biological Engineering, California Institute of Technology, Pasadena, United States; [3]Drexel University, Philadelphia, United States; [4]Department of Applied Physics, Howard Hughes Medical Institute, California Institute of Technology, Pasadena, United States

*For correspondence:
melowitz@caltech.edu (MBE);
evroth@its.caltech.edu (EVR);
kueh@uw.edu (HYK)

Present address: †Harvard Medical School, Boston, United States; ‡Cedars-Sinai Medical Center, Los Angeles, United States

Competing interests: The authors declare that no competing interests exist.

**Abstract** Cell fate decisions occur through the switch-like, irreversible activation of fate-specifying genes. These activation events are often assumed to be tightly coupled to changes in upstream transcription factors, but could also be constrained by *cis*-epigenetic mechanisms at individual gene loci. Here, we studied the activation of *Bcl11b*, which controls T-cell fate commitment. To disentangle *cis* and *trans* effects, we generated mice where two *Bcl11b* copies are tagged with distinguishable fluorescent proteins. Quantitative live microscopy of progenitors from these mice revealed that *Bcl11b* turned on after a stochastic delay averaging multiple days, which varied not only between cells but also between *Bcl11b* alleles within the same cell. Genetic perturbations, together with mathematical modeling, showed that a distal enhancer controls the rate of epigenetic activation, while a parallel Notch-dependent *trans*-acting step stimulates expression from activated loci. These results show that developmental fate transitions can be controlled by stochastic *cis*-acting events on individual loci.
DOI: https://doi.org/10.7554/eLife.37851.001

## Introduction

During development, individual cells establish and maintain stable gene expression programs through the irreversible activation of lineage-specifying regulatory genes. A fundamental goal of developmental biology is to understand how and when these activation events are initiated to drive cell fate transitions. The concentrations of active transcription factors in the nucleus are crucial for embryonic patterning and progressive gene expression changes in development (*Briscoe and Small, 2015*; *Davidson, 2010*; *Jaeger, 2011*) and are often assumed to directly dictate rates of target gene transcription (*Coulon et al., 2013*; *Estrada et al., 2016*; *Phillips, 2015*). At the same time, an additional layer of epigenetic control mechanisms acts directly at gene loci on chromosomes, through chemical modification of DNA or DNA-associated histone proteins (*Bird, 2002*; *Tessarz and Kouzarides, 2014*), or regulation of chromosome conformation or packing in the nucleus (*Felsenfeld and Dekker, 2012*). Chromatin modification and accessibility changes are ultimately initiated by the binding and action of *trans*-acting factors; however, while these changes are often assumed to closely follow transcription factor changes, other recent work shows that epigenetic processes could occur slowly (*Kaikkonen et al., 2013*; *Mayran et al., 2018*), and could introduce slow, stochastic, rate-limiting steps to gene activation, even when transcription factor inputs are fully present (*Berry et al., 2017*; *Bintu et al., 2016*). Despite much work, it has generally remained unclear

what role, if any, epigenetic mechanisms play in controlling the timing and outcome of developmental gene activation and cell fate decisions.

Epigenetic control is ordinarily difficult to disentangle from control due to changes in transcription factor activity. However, the two mechanisms can be distinguished by their effects on different gene copies in the same cell (*Bonasio et al., 2010*). Control due to transcription factor changes occurs in *trans*, and thus affects two copies of the gene in the same cell coordinately; in contrast, epigenetic mechanisms function at single gene copies, in *cis*, and thus could generate distinct activation states for different gene copies in the same cell, a concept that underlies the utility of X-chromosome inactivation and other systems as models for epigenetic gene control (*Berry et al., 2015*; *Deng et al., 2014*; *Farago et al., 2012*; *Gendrel and Heard, 2014*; *Ku et al., 2015*; *Xu et al., 2006*). For this reason, tracking both copies of a gene in the same cell with distinguishable fluorescent proteins can provide insight into the dynamics of *cis* and *trans* regulatory processes (*Elowitz et al., 2002*; *Yang et al., 2017*).

Using this approach of tracking two gene copies, we have studied the developmental activation of *Bcl11b*, a key driver of T-cell commitment and identity. To become a T-cell, hematopoietic progenitors transition through a series of developmental states, where they lose alternate lineage potential and eventually commit to the T-cell lineage (*Figure 1A*). T-cell lineage commitment requires the irreversible switch-like activation of *Bcl11b*, which serves to repress alternate lineage potential and establish T-lineage identity (*Ikawa et al., 2010*; *Li et al., 2010a*; *Li et al., 2010b*). *Bcl11b* is regulated by an ensemble of transcription factors, including Runx1, GATA-3, TCF-1, and Notch, which bind to multiple locations on the gene locus (*Li et al., 2013*; *Kueh et al., 2016*). However, even when these developmentally controlled transcription factors have been fully mobilized, *Bcl11b* activation occurs only after an extended time delay of ~4 days, allowing pre-commitment expansion of progenitors (*Kueh et al., 2016*). During activation, the *Bcl11b* locus remodels its epigenetic state, undergoing changes in DNA methylation and histone modification (*Ji et al., 2010*; *Zhang et al., 2012*), nuclear positioning, genome compartmentalization and looping interactions (*Hu et al., 2018*), and expression of a *cis*-acting lncRNA transcript (*Isoda et al., 2017*). These observations suggest that the dynamics of *Bcl11b* activation could be determined by epigenetic processes as well as transcription factors.

To separately follow two *Bcl11b* copies in developing cells, we engineered a dual-color reporter mouse, where the two *Bcl11b* copies are tagged with distinguishable fluorescent proteins. We then used quantitative live-cell imaging to follow *Bcl11b* activation dynamics in single progenitor lineages, along with mathematical modeling and perturbation experiments to dissect the relative contributions of *cis*- and *trans*- acting inputs to *Bcl11b* regulation. Our results revealed that activation of *Bcl11b* and consequent T-cell commitment require a stochastic, *cis*-acting epigenetic step on the *Bcl11b* locus. This step occurs independently at the two alleles in the same cell, with a slow timescale spanning multiple days and cell cycles. A separate *trans*-acting step, controlled by the T-cell developmental signal Notch, occurs in parallel with this *cis*-acting step and provides an additional necessary input for *Bcl11b* activation. Finally, we found that over the course of development, T-cell progenitors lose the ability to activate the *cis*-epigenetic switch, and as a result, can progress to final differentiated states with only one *Bcl11b* locus stably activated. Together, these results show that intrinsically stochastic events occurring at single gene copies can determine the timing and outcome of mammalian cell fate decisions.

## Results

### Two Bcl11b copies show slow, independent activation in single progenitor lineages

We generated a double knock-in reporter mouse strain, with an mCitrine yellow fluorescent protein (YFP) inserted non-disruptively in the 3'-untranslated region of one *Bcl11b* copy and an mCherry red fluorescent protein (mCh) at the same site in the other copy (*Figure 1B* and *Figure 1—figure supplement 1*). Both *Bcl11b* copies contain a floxed neomycin resistance cassette downstream of the fluorescent protein (*Figure 1—figure supplement 1*); however, we have shown conclusively, using Cre-mediated excision, that this cassette has no effect on *Bcl11b* activation (*Kueh et al., 2016*). We isolated thymocyte populations at different stages of development and differentiation directly from

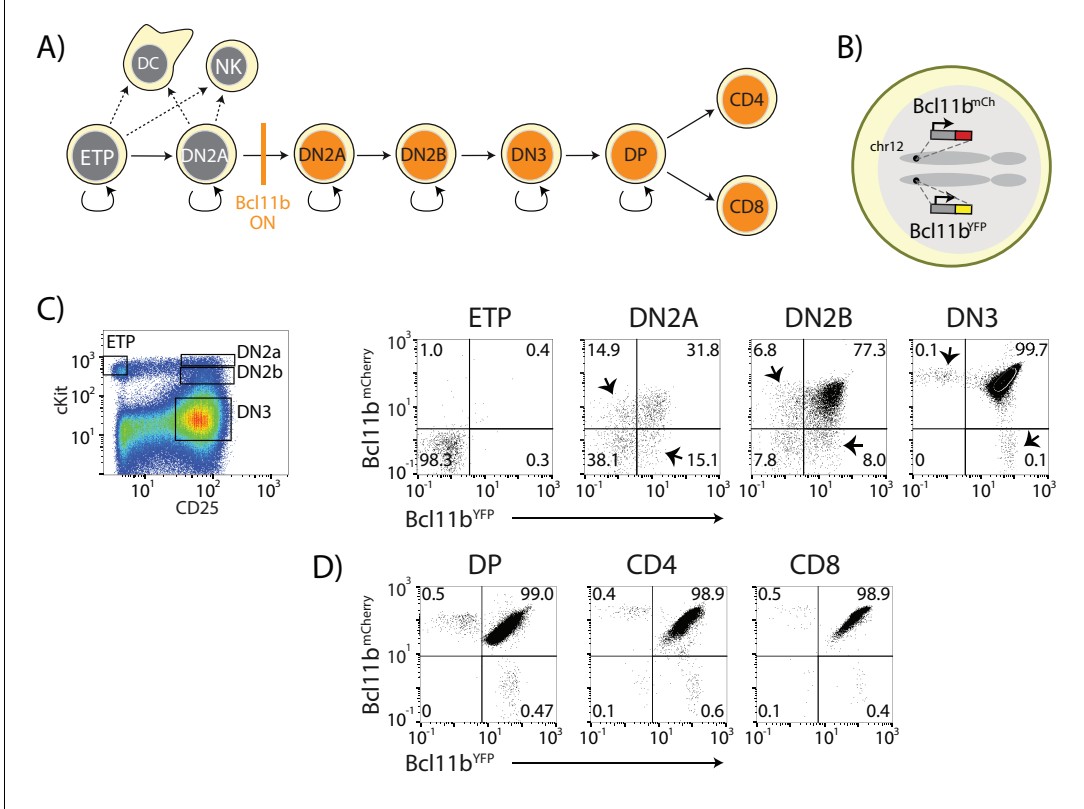

**Figure 1.** Dual-color *Bcl11b* reporter strategy can reveal epigenetic mechanisms controlling T-cell lineage commitment. (A) Overview of early T-cell development. *Bcl11b* turns on to silence alternate fate potentials and drive T-cell fate commitment. ETP – early thymic progenitor; DN2 – CD4⁻CD8⁻double negative-2A progenitor; DP – CD4⁺ CD8⁺; NK – natural killer; DC – dendritic cell. (B) Dual-allelic *Bcl11b* reporter cells, where two distinguishable fluorescent proteins (YFP and mCherry) are inserted non-disruptively into the same sites on the two endogenous *Bcl11b* loci. (C) Flow cytometry plots show cKit versus CD25 levels in CD4⁻CD8⁻ double negative (DN) thymic progenitors (left), along with Bcl11b-YFP versus Bcl11b-mCh expression levels in the indicated DN progenitor subsets from dual *Bcl11b* reporter mice. Arrowheads indicate cells expressing one copy of *Bcl11b*. (D) Flow plots show Bcl11b-YFP versus Bcl11b-mCh levels in CD4⁺CD8⁺double positive (DP) T-cell precursors from the thymus (left), or CD4 (center) or CD8 (right) T-cells from the spleen. Results are representative of analysis of 6–8 mice from two independent experiments. See also *Figure 1—figure supplement 1*.

DOI: https://doi.org/10.7554/eLife.37851.002

The following figure supplement is available for figure 1:

**Figure supplement 1.** Experimental strategy for generating different *Bcl11b* reporter mouse strains.

DOI: https://doi.org/10.7554/eLife.37851.003

dual reporter *Bcl11b* mice, and measured the fraction of cells expressing *Bcl11b* from each allele at stages spanning the initial onset of *Bcl11b* expression (*Figure 1C*). As reported previously (*Kueh et al., 2016*; *Tydell et al., 2007*), *Bcl11b* was inactive in early T-cell progenitors (ETPs), and began to turn on in the subsequent CD4, CD8 double negative (DN)2a stage, becoming expressed in all cells throughout the rest of T-cell development (*Figure 1A,C,D*). By DN2b and DN3 stages, the large majority of cells had turned on both *Bcl11b* copies. These transitions involve multiple cell cycles each, with about 2 days between late ETP and DN2a and about 3 days between DN2a and DN2b (*Kueh et al., 2016*). However, in the DN2a compartment where *Bcl11b* gene activation begins, a significant fraction of the cells expressed only one copy of *Bcl11b*, with roughly equal fractions of cells expressing either the YFP or mCherry (mCh) alleles (*Figure 1C*, arrowheads). This suggested that the two *Bcl11b* copies could turn on at different times in the same cell during development.

To determine directly whether two *Bcl11b* copies switch on independently in the same cell, we used multi-day timelapse imaging to follow the two *Bcl11b* fluorescent reporters in clonal lineages of developing progenitors. We isolated *Bcl11b*-negative DN2a T-cell progenitors from dual *Bcl11b*

reporter mice, transduced them with cyan fluorescent protein (CFP)-expressing retroviral constructs for cell tracking, and cultured them *in vitro* at limiting dilution with OP9-DL1 stromal cells (*Schmitt and Zúñiga-Pflücker, 2002*), confining them in microwells to allow tracking of descendants of each progenitor over multiple days (*Figure 2A*). We used OP9-DL1 cells to present the Notch ligand DL1, a critical T-cell developmental signal, and included the supportive cytokines Interleukin (IL)−7 and Flt3 ligand (see Materials and methods). The ~1 hr interval between successive frames did not permit complete lineage tracking due to rapid cell movement (*Video 1*), but still enabled mapping and visualization of all descendants, and determination of coarse-grained lineage relationships (*Figure 2B,C*, bottom left).

We had previously shown that about 3 days are required for half of the cells in such DN2a populations to turn on any Bcl11b expression (*Kueh et al., 2016*). In theory, this delay could reflect requirement for activation of some additional transcription factor. However, even a novel transcription factor would be able to work on both alleles in parallel. Instead, strikingly, imaging revealed strongly asynchronous activation of the two *Bcl11b* copies in the same cell during this time period. Within single clonal lineages from DN2a progenitors, one copy of *Bcl11b* could switch on multiple days and cell generations before the other (*Figure 2B,C* and *Video 1*), giving rise to distinct allelic expression states that persisted over multiple divisions. Across clones, however, similar percentages of cells activated *Bcl11b-YFP* first as compared to those turning on *Bcl11b-mCherry* first, consistent with independent activation (*Figure 2D,E*). Similar activation rates for both alleles were also observed on average across populations from individual mice regardless of parent-of-origin (data not shown), ruling out any imprinting-type bias. These results indicated that the allelic bias within clones was clonally inherited.

From a dynamic point of view, we observed that in some clones, the times at which a *Bcl11b* allele first turned on differed between progeny of a single cell (*Figure 2C*, 42 hr), such that individual progenitors frequently gave rise to clonal descendants with multiple distinct states of *Bcl11b* allelic activation (53.3% heterogeneous after 4d, N = 15, *Figure 2—figure supplement 1*). A substantial percentage (~40%) of all cells remained mono-allelic in expression after 4d (*Figure 2E*), indicating that stochastic locus activation occurs with a slow time constant spanning multiple days. Furthermore, the fractions of cells mono-allelically expressing Bcl11b-YFP or Bcl11b-mCh increased with the same dynamics, indicating that each locus is triggered with the same stochastic activation rate. We note that the percentages of mono-allelic cells generated at given timepoints on OP9-DL1 co-culture differed from those in DN2b progenitors from the thymus, which have emerged from Bcl11b non-expressing DN2a cells at some unknown time in the past (40% versus ~15%, *Figure 1C*). This could reflect differences in the kinetics of *Bcl11b* allelic activation between the OP9-DL1 system and the thymic microenvironment. However, in both cases, we observed clearly defined *Bcl11b* mono-allelic as well as bi-allelic expressing populations, implying that the same slow *cis-* event observed in *in vitro* tracking experiments also governs *Bcl11b* activation and T-lineage commitment in the thymus. Taken together, these results suggest that timing of the *Bcl11b* activation switch – and the ensuing commitment to become a T-cell – is controlled independently at each *Bcl11b* allele by a stochastic and remarkably slow rate-limiting step.

## A distal enhancer modulates stochastic Bcl11b locus activation

The stochastic transition of *Bcl11b* from an inactive to active state may be controlled by specific *cis*-regulatory DNA elements on the *Bcl11b* locus. Consistent with this idea, we found that graded changes in Notch signaling, GATA-3 activity, and TCF-1 activity alter the likelihood of all-or-none activation, rather than the amplitude of transcription (*Kueh et al., 2016*). Indeed, in a number of systems, *cis*-regulatory elements do not appear to control transcriptional amplitudes, but instead modulate the probabilities of all-or-none activation (*Fukaya et al., 2016*; *Khan et al., 2011*; *Walters et al., 1995*; *Weintraub, 1988*). To test how stochastic activation of individual *Bcl11b* alleles may be controlled, we examined the effect of disrupting the one known positive *cis*-regulatory element region, which resides ~850 kb downstream of *Bcl11b* within a 'super enhancer' at the opposite end of the same topologically associated domain (*Li et al., 2013*) (*Figure 3A*). This region, which shows distinctive histone marking and some T-lineage-specific transcription factor occupancy even before *Bcl11b* activation (*Kueh et al., 2016*), lies about 11 kb from the promoter of a *Bcl11b*-associated lncRNA, and loops to the *Bcl11b* gene body in a T-cell lineage specific manner (*Hu et al.,*

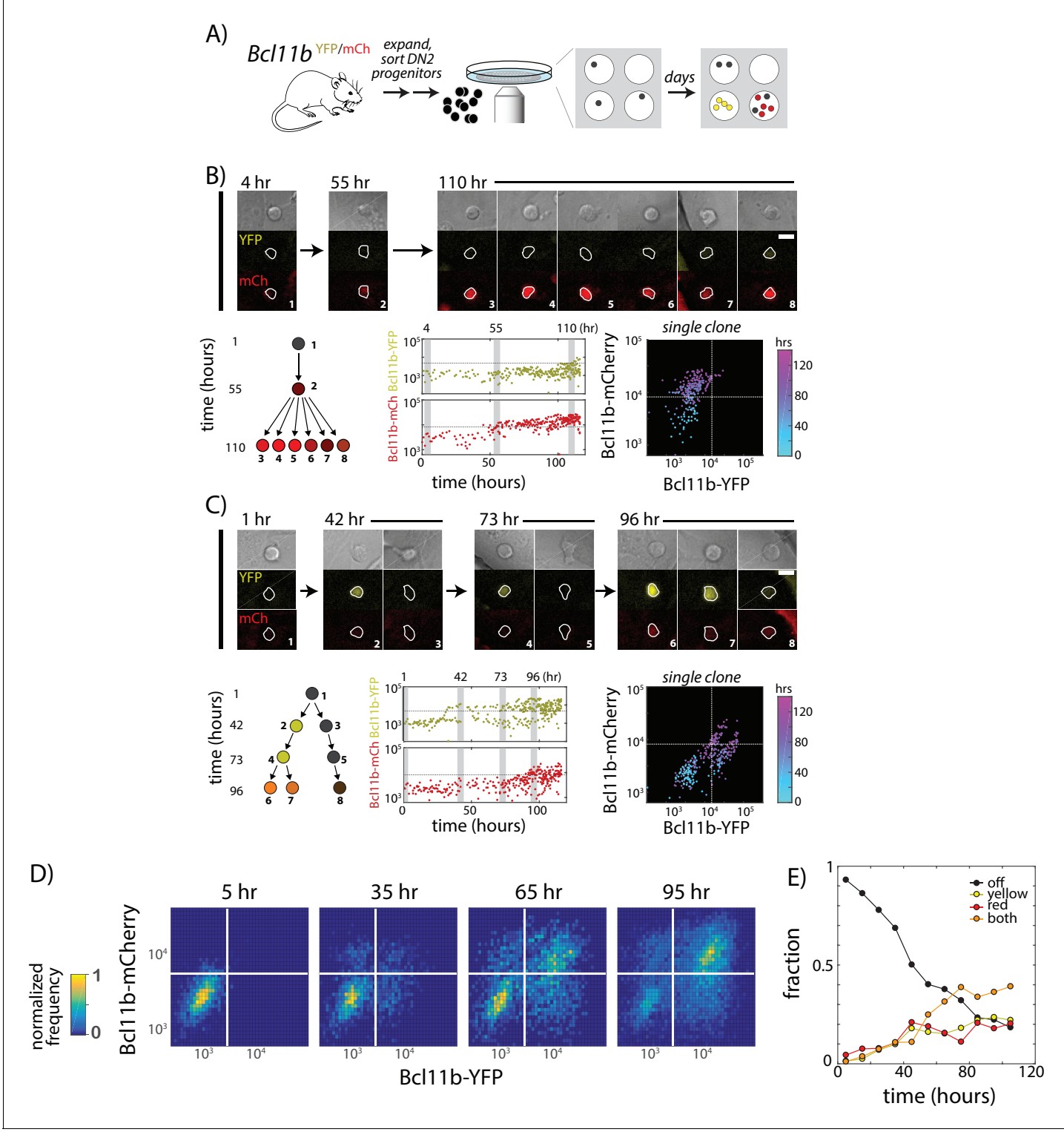

**Figure 2.** Two copies of *Bcl11b* switch on independently and stochastically in the same cell in single lineages of T-cell progenitors. (**A**) *Bcl11b*-negative DN2 cells derived from bone-marrow progenitors were isolated by flow cytometry, cultured within microwells, and followed for 5 days using fluorescence imaging. Cells were then segmented using automated image analysis. (**B–C**). Dynamics of *Bcl11b* activation in two representative clonal progenitor lineages. Timelapse images (top) show developing T-cell progenitors from two representative clones (left), with segmented cell boundaries in white. Numbers (top left) indicate time in hours. Scale bar = 10 microns. Trees (bottom left) show coarse-grained cell lineage relationships for the cells shown here. Plots (center, lower rows) show Bcl11b-YFP and Bcl11b-mCh expression time traces in all cells from a single clone, with vertical gray bars indicating the time points of the image shown on the left. Horizontal lines indicate activation threshold. Colored scatterplots (bottom right) show

*Figure 2 continued on next page*

*Figure 2 continued*

time evolution of Bcl11b-mCh versus Bcl11b-YFP levels in single clones, from 0 hr (cyan) to 120 hr (purple). (D) Heat maps show Bcl11b-YFP and Bcl11b-mCh distributions in the polyclonal population at the indicated time points. White lines represent *Bcl11b* expression thresholds. Color bar (left) represents normalized cell numbers at each time point. (E) Fractions of cells having different *Bcl11b* allelic expression states over time, obtained by mixed Gaussian fitting of the heat maps shown. Data represent a cohort of ~200 starting cells from a single timelapse movie. Overall, data show that *Bcl11b* switches on slowly and stochastically in single lineages of progenitors, maintaining alternate activity states in the same clone, heritable across many divisions. Results are representative of three independent experiments. See also *Figure 2—figure supplement 1* and *Video 1*.

DOI: https://doi.org/10.7554/eLife.37851.004

The following source data and figure supplement are available for figure 2:

**Source data 1.** Differential *Bcl11b* allelic expression states over time for a cohort of ~200 starting cells.
DOI: https://doi.org/10.7554/eLife.37851.006

**Figure supplement 1.** *Bcl11b* shows heterogeneity in locus activation within clonal progenitor lineages.
DOI: https://doi.org/10.7554/eLife.37851.005

*2018*; *Isoda et al., 2017*; *Li et al., 2013*). Like the *Bcl11b* locus itself, this enhancer region is marked by H3K27me3 in non-T lineage cells (*Li et al., 2013*).

Using standard gene targeting, we deleted this distal ~2 kb enhancer region on the *Bcl11b-YFP* allele, leaving the *Bcl11b-mCherry* allele intact, to generate *Bcl11b*$^{YFP\Delta Enh/mCh}$ dual reporter mice (*Figure 3A* and *Figure 1—figure supplement 1*), and then analyzed resultant effects on YFP regulation in different T-cell subsets (*Figure 3B,C* and *Figure 3—figure supplements 1–3*). These were analyzed either from established young adult *Bcl11b*$^{YFP\Delta Enh/mCh}$ mice (*Figure 3B,C* and *Figure 3—figure supplements 1,2*) or from adult chimeras populated with fetal liver cells from the $F_0$ generation (*Figure 3—figure supplement 3A,B*). The non-disrupted *Bcl11b-mCherry* allele served as an internal, same-cell control. At the ETP stage, essentially all *Bcl11b* alleles were silent, regardless of whether they had an intact or disrupted enhancer, as expected (*Figure 3B*). During the DN2a and DN2b stages, the enhancer-disrupted *Bcl11b-YFP* allele showed dramatically reduced activation compared to the *Bcl11b-mCherry* allele in the same cell. Interestingly, at later developmental stages in the thymus and in peripheral T-cell subsets (CD4, Treg, CD8) a large fraction of cells showed expression of the enhancer-disrupted *YFP* allele, along with the wild-type *Bcl11b-mCherry* allele (*Figure 3B,C* and *Figure 3—figure supplements 1,2,3*), indicating that the targeted element is not indispensable for *Bcl11b* activation. However, a small but significant percentage of cells still failed to activate the enhancer-disrupted allele, and instead persisted in a mono-allelic state with only expression of the *Bcl11b-mCherry* allele (*Figure 3B,C* and *Figure 3—figure supplements 1,2,3*). Mono-allelic cells were found in memory as well as naïve T-cell subsets (*Figure 3—figure supplement 2*), implying that these mono-allelically expressing cells are capable of immune responses, as expected from the normal phenotype of *Bcl11b* knockout heterozygotes. As shown in fetal liver chimeras, generation and persistence of Bcl11b-mCherry mono-allelic cells due to the mutant *Bcl11b*$^{YFP\Delta Enh}$ allele were determined cell intrinsically (*Figure 3—figure supplement 3*). However, from flow cytometric profiles, cells that turned on the disrupted allele expressed it at normal levels, suggesting that the enhancer mutation reduced the stochastic rate of *Bcl11b*

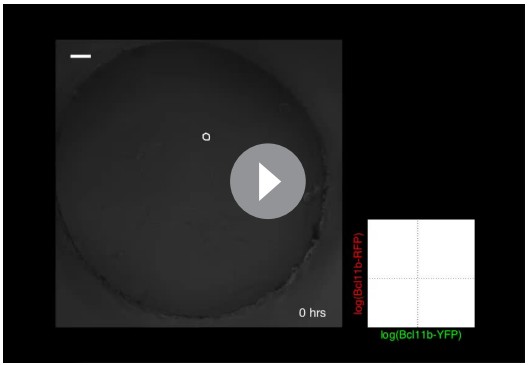

**Video 1.** Timelapse movie of a single clonal DN2 progenitor lineage. Bcl11b-YFP⁻mCh⁻ DN2 progenitors were cultured on OP9-DL1 monolayers with 5 ng/mL IL-7 and Flt3-L within individual PDMS micro-wells, and continuously imaged for 100 hr. Images show superposition of a DIC image (gray) and cellular fluorescent intensities from the Bcl11b-mCherry (red) and Bcl11b-YFP (green) channels, with segmented cell boundaries shown in white. For clarity, images show only the fluorescence intensities within the cell boundaries, excluding auto-fluorescence from well boundaries and OP9-DL1 monolayers. Scatter-plot (bottom-right) updates with each frame to show fluorescent intensities of segmented cells at corresponding time points. Scale bar = 50 microns.
DOI: https://doi.org/10.7554/eLife.37851.007

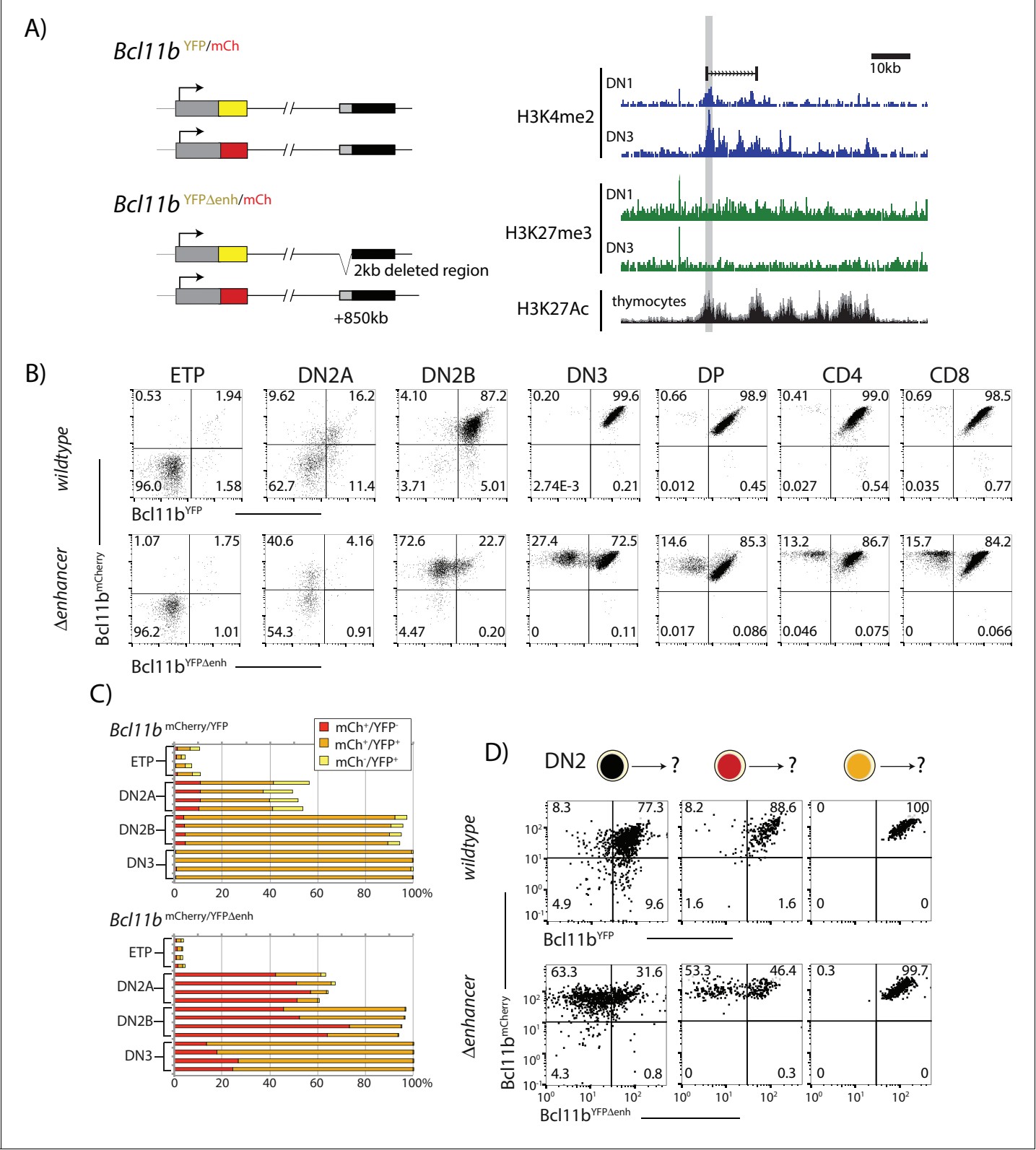

**Figure 3.** A distal enhancer region controls *Bcl11b* activation probability. (**A**) Schematic of normal and enhancer-deleted two-color *Bcl11b* reporter strains (left). Genome browser plots (right), showing +850 kb enhancer of *Bcl11b*, showing distributions of histone marks (H3K4me2, H3K27me3, and H3K27Ac) and an associated LncRNA (*Isoda et al., 2017*). Orientation is with transcription from left to right (reversed relative to genome numbering). Gray shaded area indicates the enhancer region deleted using gene targeting (removed region: chr12:108,396,825–108,398,672, mm9). (**B**) Flow

*Figure 3 continued on next page*

*Figure 3 continued*

cytometry plots show Bcl11b-mCh versus Bcl11b-YFP levels in developing T-cell populations from dual *Bcl11b* reporter mice, either with an intact YFP enhancer (top), or a disrupted YFP enhancer (bottom). Results are representative of two independent experiments. (C) Bar graphs showing the percentages of cells in early thymic populations with mono- and bi-allelic expression of wildtype mCherry and wildtype YFP versus mutant YFP alleles in wildtype *Bcl11b*[YFP/mCh] and *Bcl11b*[YFPΔEnh/mCh] dual reporter mice, demonstrating the reduced frequency of mutant YFP allele expression relative to the wildtype mCherry allele in the same cells. Each bar shows results from one mouse; n = 4 mice of each strain are shown. (D) DN2 progenitors were sorted for different *Bcl11b* allelic activation states as indicated, cultured on OP9-DL1 monolayers for 4 days, and analyzed using flow cytometry. Flow plots show Bcl11b-mCh versus Bcl11b-YFP levels of cells generated from precursors with a normal (top) or disrupted (bottom) YFP enhancer, showing defective YFP up-regulation from the mutant relative to the wildtype alleles. Enhancer disruption reduces the probability of switch-like *Bcl11b* activation, but does not affect expression levels after activation. Results are representative of two independent experiments. See also *Figure 1—figure supplement 1*, and *Figure 3—figure supplement 1–4*.

DOI: https://doi.org/10.7554/eLife.37851.008

The following source data and figure supplements are available for figure 3:

**Source data 1.** Comparison of *Bcl11b* allelic expression between wildtype and mutant dual reporter mice in early thymic populations.
DOI: https://doi.org/10.7554/eLife.37851.016

**Figure supplement 1.** Levels of mono-allelic *Bcl11b* expression in thymus subsets: mono-allelic expression can persist throughout thymic development.
DOI: https://doi.org/10.7554/eLife.37851.009

**Figure supplement 1—source data 1.** Percentages of mono- and bi-allelic expressing cells in specific thymic populations analyzed for wildtype (Bcl11b[YFP/mCh(neo)]) and mutant (Bcl11b[YFPΔEnh/mCh(neo)]) dual reporter mice.
DOI: https://doi.org/10.7554/eLife.37851.010

**Figure supplement 2.** Mono-allelic *Bcl11b* expression persists in peripheral splenic T-cell subsets and is cell autonomous.
DOI: https://doi.org/10.7554/eLife.37851.011

**Figure supplement 2—source data 1.** Percentages of mono- and bi-allelic expressing cells in specific spleen populations analyzed for wildtype (Bcl11b[YFP/mCh(neo)]) and mutant (Bcl11b[YFPΔEnh/mCh(neo)]) dual reporter mice.
DOI: https://doi.org/10.7554/eLife.37851.012

**Figure supplement 3.** Cell autonomy of Bcl11b expression control in hematopoietic chimeric mice.
DOI: https://doi.org/10.7554/eLife.37851.013

**Figure supplement 3—source data 1.** Percentages of mono- and bi-allelic expressing cells in thymic and splenic populations analyzed for wildtype (Bcl11b[YFP/mCh(neo)]) and mutant (Bcl11b[YFPΔEnh/mCh(neo)]) chimeric mice.
DOI: https://doi.org/10.7554/eLife.37851.014

**Figure supplement 4.** Thymocytes from homozygous mutant enhancer Bcl11b[YFPΔEnh/YFPΔEnh] mice are able to generate T-cell subsets expressing Bcl11b at normal levels relative to wild-type enhancer Bcl11b YFP/YFP mice.
DOI: https://doi.org/10.7554/eLife.37851.015

activation, but not its expression level once activated.

To directly test this hypothesis, we measured *Bcl11b-YFP* activation with or without enhancer disruption, by sorting DN2 progenitors with zero or one allele activated, culturing on OP9-DL1 feeders, and analyzing activation dynamics of both alleles using flow cytometry. Consistently, enhancer disruption greatly reduced the fraction of cells that turned on *Bcl11b-YFP*, but did not perturb its expression level in cells that already successfully activated it (*Figure 3D*). Neither the wildtype nor the enhancer-disrupted allele reverted to silence after being activated. These results show that the deleted region within the distal *Bcl11b* super-enhancer works selectively, in *cis*, to accelerate the irreversible stochastic switch of the *Bcl11b* locus from an inactive to an active state.

The activation of the enhancer-disrupted *Bcl11b* allele observed in many DN2b and later cells suggests that there are other *cis*-regulatory elements on the *Bcl11b* locus that can also promote stochastic locus activation. The extended intergenic gene desert between *Bcl11b* and the next gene, *Vrk1*, is rich in potential regulatory elements that could compensate for the disruption of the enhancer element in the cells activating the YFP allele (*Hu et al., 2018*). Alternatively, the intact enhancer at the mCherry-tagged locus in the same cell could activate the enhancer-deleted *Bcl11b* locus in trans, but this was ruled out when we bred mice with the enhancer disruption to homozygosity (Bcl11b[YFPΔEnh/YFPΔEnh]). Progenitors from these mice were still able to turn on *Bcl11b* and to undergo T-cell development to CD4, CD8 double positive (DP) and single positive (SP) cells, and all the cells in these populations had normal levels of *Bcl11b* expression (*Figure 3—figure supplement 4*). Thus, the enhancer we identified works together with other regulatory elements specifically to control *Bcl11b* activation timing.

# A parallel *trans*-acting step enables expression from an activated Bcl11b locus

The known transcriptional regulators of *Bcl11b*—TCF-1, Gata3, Notch1 and Runx1—are already strongly expressed prior to entering the DN2 stage, suggesting they are not limiting for *Bcl11b* activation in DN2 cells. The data presented above show that *cis*-acting mechanisms can substantially slow activation at individual alleles. However, additional *trans*-acting factors or post-translational changes in these factors could still limit the kinetics of *Bcl11b* activation, working either upstream of the *cis*-opening mechanism or as a separate, independent requirement. To gain insight into whether such *trans*-acting inputs are necessary to explain the observed dynamics, and how they could act together with the *cis*-acting step, we developed a set of minimal models requiring the *cis*-activating step either alone or together with an additional *trans*-acting step (see Appendix for model details). While the analysis is subject to technical limitations such as fluorescent protein sensitivity and time delays in expression, and the actual biological behavior is undoubtedly more complex, these minimal models, nevertheless, enable discrimination among broad classes of behavior.

To obtain unbiased estimates of the population fractions from imaging data (*Figure 2*), we fit four two-dimensional Gaussians to single-cell Bcl11b-YFP and Bcl11b-mCherry data (see Materials and methods for details). We note that estimated population fractions increased after a delay (*Figure 4B*, gray-shaded area), due to a time lag in fluorescence accumulation and detection after activation (*Figure 4—figure supplement 1*). We accounted for this lag by incorporating a fixed time delay into all of our model fits (see Materials and methods).

In the simplest '*cis* only' model, we assume that only the *cis*-activation step is required for *Bcl11b* activation in DN2 stage, with all required *trans*-acting steps having occurred prior to the ETP-DN2 transition (*Figure 4A*, left). Because *cis*-activation is controlled at each allele by a single rate constant, this model predicts a substantial lag between the appearance of mono-allelic cells, which require one *cis*-activation event, and the appearance of bi-allelic cells, which require two independent events (Appendix). By contrast, in experiments, bi-allelic cells accumulated immediately (*Figure 4B* and *Figure 4—figure supplement 1*), without a substantial lag relative to mono-allelic ones, resulting in a poor fit of the data to the *cis*-only model (*Figure 4A*). These results rule out the simplest *cis*-only model, and suggest that additional *trans* events may still limit *Bcl11b* expression at the DN2 stage.

We next considered two models in which *trans*-acting events affect *Bcl11b* activation. In the 'sequential *trans*-*cis*' model, a *trans* step must occur prior to the *cis*-activation step (*Figure 4A*). This *trans* step could represent activation of a factor or epigenetic regulator that is necessary for *cis*-activation. In the 'parallel *trans*-*cis*' model, both *cis* and *trans* steps are similarly necessary, but can occur in either order (*Figure 4A*). In this case, the *trans* step could represent activation of a factor that drives *Bcl11b* transcription, but only from a *cis*-activated locus. While our models only consider the DN2 stage, we note that they allow for some events to occur prior to the ETP-DN2a transition (*Figure 4A*, gray dotted arrows). When the *trans*-acting step is rate limiting, both of these models reduce bi-allelic lag by allowing the two alleles to turn on in relatively quick succession (in either model) or simultaneously (in the parallel model). For this reason, both the sequential and parallel *trans*-*cis* models reduced the lag prior to accumulation of bi-allelic cells, and hence fit the data significantly better than the '*cis* only' model (*Figure 4B*, p<0.01 for both models).

While the sequential and parallel models show similar bulk behavior, they make divergent predictions about the distributions of mono-allelic and bi-allelic expression states within clonal lineages. For example, in the sequential model, silent progenitors are equally likely to activate one or the other *Bcl11b* allele, and are thus more likely to show mono-allelic expression from both alleles in single clones (*Figure 4D*, 'mixed mono-allelic'). In contrast, in the parallel model, non-expressing progenitors could have one *cis*-activated but unexpressed *Bcl11b* allele due to absence of the *trans* step. Clonal descendants of such cells would be predisposed to show mono-allelic expression from the same allele before activating the second (*Figure 4D*, 'single mono-allelic'). Therefore, to discriminate between sequential and parallel activation models, we used Monte-Carlo methods to simulate the dynamics of *Bcl11b* activation in all descendants of a single starting cell over four generations for each of the two models (Appendix), using the parameters that gave the best fits to the global time course data in *Figure 4B*. Altogether, we generated and analyzed N = 30,000 clonal lineages for each model.

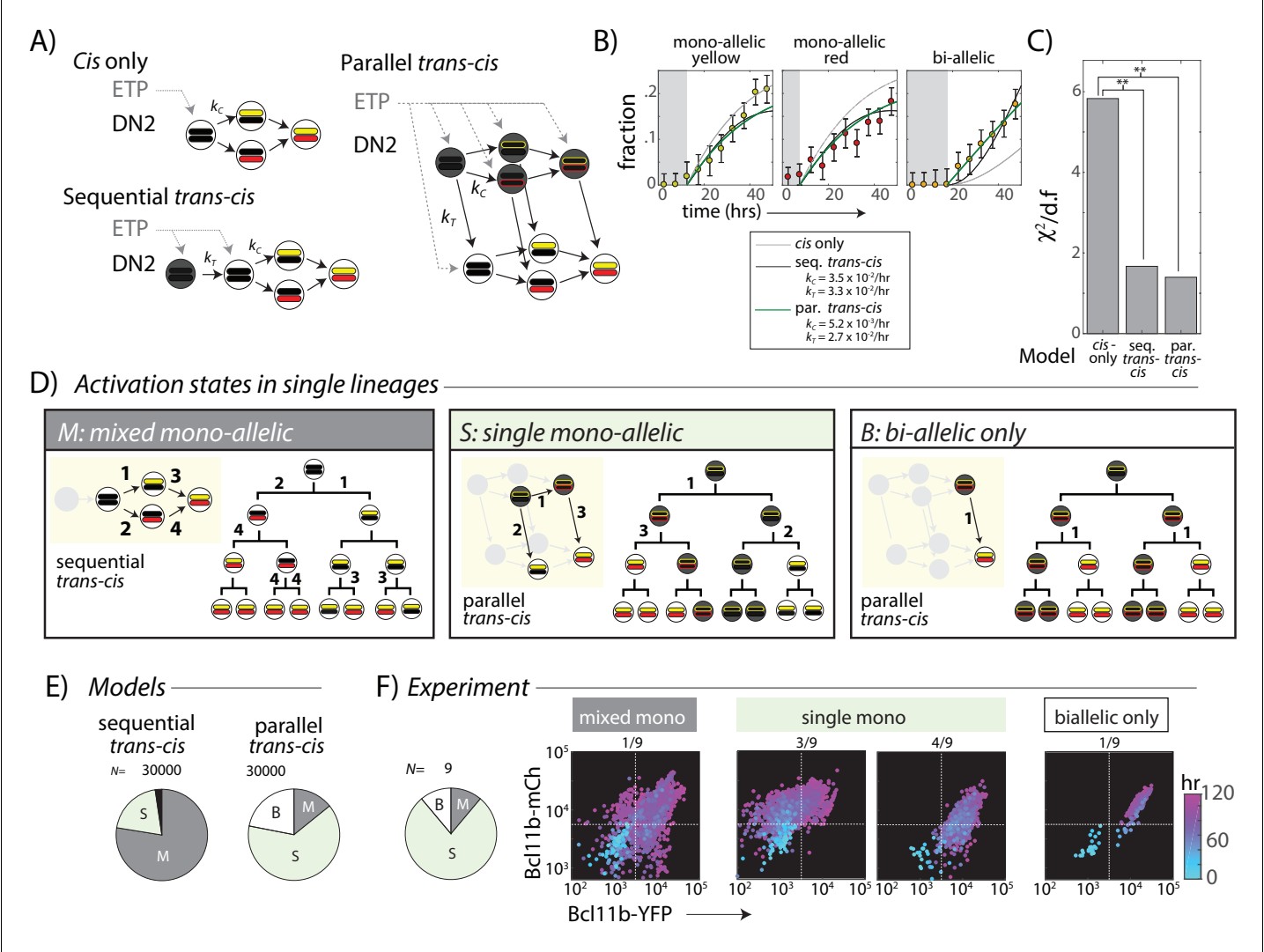

**Figure 4.** A *trans*-acting step, occurring in parallel with the *cis*-acting step, provides an additional input for *Bcl11b* activation. (**A**) Candidate models for *Bcl11b* activation from the DN2 stage, involving a single *cis*-acting switch (top left), sequential *trans*-, then *cis*-acting switches (bottom left), and parallel, independent *trans*- and *cis*- acting switches (right). (**B**) Plots show best fits of different models to the time evolution of *Bcl11b* allelic activation states, observed by timelapse imaging (*Figure 2*). Gray-shaded area indicates time delay for detection of indicated allelic state as a result of the time required for stable fluorescence protein accumulation. Best fit rate constants indicated in legend. (**C**) Bar charts show reduced chi-squared values for each model fit, that is the normalized sum-squared fit errors over all time points and allelic states, divided by the degrees of freedom (d.f.) (see Materials and methods). Both sequential and parallel *trans-cis* models fit the data significantly better than the *cis*-only model (F-test, $F = 12.2$, p=0.0052, sequential vs. *cis*-only model; $F = 8.13$, p=0.021, parallel vs. *cis*-only model). (**D**) Three possible classes of *Bcl11b* activation states observable from clonal lineage data. Lineage trees and transition diagrams show examples of simulated lineages that fall into the indicated classes. (**E**) Pie charts show expected distribution of allelic activation states predicted for clonal lineages of non-expressing progenitors in either the sequential (left) or the parallel (right) *trans-cis* model, obtained from $N = 30,000$ simulations, using parameters derived from bulk fitting (see Appendix). (**F**) Pie chart (left) shows observed distribution of activation states observed across an entire imaging time course. Colored scatterplots (right) show Bcl11b-mCh versus Bcl11b-YFP levels of single-cell lineages, falling into the indicated categories. Clones were scored according to observable fluorescence across an entire developmental trajectory, from 0 hr (cyan) to 120 hr (purple). The observed frequency of clones with 'single mono-allelic' expression of *Bcl11b* (7/9 = 77%) is significantly different than that predicted for the sequential *trans-cis* Model (20.1%, **- p<0.001, $\chi^2 = 14.9$, d.f. = 1), but not significantly different from that predicted for the parallel *trans-cis* Model (63.9%, $\chi^2 = 0.27$, d..f = 1, n.s.). Results are representative of three independent experiments. See *Figure 4—figure supplement 2* for data for independent replicate experiments.

DOI: https://doi.org/10.7554/eLife.37851.017

The following source data and figure supplements are available for figure 4:

**Source data 1.** Quantitative analysis of timelapse imaging data used to test three minimal models.

DOI: https://doi.org/10.7554/eLife.37851.020

*Figure 4 continued on next page*

*Figure 4 continued*

**Figure supplement 1.** Least-squares fitting of 2D histograms of Bcl11b expression levels.

DOI: https://doi.org/10.7554/eLife.37851.018

**Figure supplement 2.** Clones show mono-allelic expression from a single predominant allele during *Bcl11b* activation.

DOI: https://doi.org/10.7554/eLife.37851.019

As intuitively expected, the sequential *trans-cis* model predominantly generated 'mixed mono-allelic' clones containing cells with mono-allelic expression of both alleles, with or without bi-allelically expressing cells (*Figure 4D,E*, 'mixed mono-allelic'). These distributions reflect the most likely event trajectory in the sequential model, in which independent, unsynchronized *cis*-activation events occur at each *Bcl11b* locus in different cells from a single ancestor. Within a cohort of clonal descendants competent to activate the *cis*-step, the first-activated allele choice occurs independently in each cell, generating multiple paths towards bi-allelic *Bcl11b* activation within a single clone. By contrast, the parallel model generated a much smaller fraction of such 'mixed mono-allelic' clones, and predominantly generated clones in which mono-allelic expression was restricted to the same allele across most cells (*Figure 4D,E*, 'single mono-allelic'). This intra-clonal bias arises when the *cis*-acting step at one locus precedes the *trans*-step, forcing still-non-expressing DN2a precursors to preferentially activate that locus once the *trans*-acting event occurs (*Figure 4D*). Because the rate of *cis*-activation is low ($\tau_c \sim$ 4–6 days in this model), multiple individual cells can be generated within a clone that have inherited the same activated allele of the locus, prior to full bi-allelic expression. Moreover, the parallel but not the sequential *trans-cis* model gave rise to a small fraction of clones that showed only bi-allelic expression (*Figure 4D,E*, 'bi-allelic only'), reflecting the activation of the *trans*-limiting step in cells that had already undergone *cis*-activation of both *Bcl11b* copies.

To discriminate experimentally between these two models, we quantified the distribution of *Bcl11b* allelic activation states generated in clonal lineages from progenitors starting with no *Bcl11b* activation, observed by timelapse microscopy as described above (*Figure 2*). Within a clone, we most frequently observed mono-allelic expression from only one specific allele, with or without bi-allelically expressing cells (*Figure 4F* 'single mono-allelic', light green, 7/9 clones, *Figure 4—figure supplement 2*; similar results observed over three independent experiments), but only rarely observed mono-allelic expression from both loci within the same clone (*Figure 4F*, 'mixed mono-allelic', gray, 1/9 clones). The observed percentage of 'single mono-allelic' expressing clones (7/9 = 77%) was significantly greater than that expected from the sequential *trans-cis* model (20.4%, p<0.005). Moreover, in one clone, we observed concurrent activation of both alleles (*Figure 4F*), a behavior that would have been exceedingly rare in a sequential model (none observed in 30,000 simulations). Together, these results suggest that a *trans*-acting step, acting in parallel with the *cis*-acting step, controls *Bcl11b* expression.

## Notch signaling controls the parallel trans-acting step in Bcl11b activation

Notch signaling drives T-cell fate commitment and provides an important input for *Bcl11b* expression. While not required to maintain *Bcl11b* expression in committed cells, it acts earlier to enhance the probability of all-or-none *Bcl11b* expression at the DN2 stage and stabilize *Bcl11b* expression shortly after activation, preventing the re-silencing that still can occur in a small fraction of newly expressing cells (*Kueh et al., 2016*). The Notch intracellular domain is diffusible in the nucleus, but could affect *Bcl11b* activation by modulating either the *cis* or *trans* step in the parallel model. For example, Notch signaling could activate a *trans* factor that drives *Bcl11b* transcription from a *cis*-activated locus, and thereby alter the fraction of cells that express *Bcl1b* from a *cis*-activated locus. Alternatively, Notch could affect the rate of the *cis*-activation process, for instance by enhancing the activity of chromatin-remodeling enzymes on the *Bcl11b* promoter or enhancer, in which case it might alter the ratio of mono-allelic to bi-allelic activation states. To distinguish between these cases, we first experimentally analyzed the effects of removing Notch signaling on *Bcl1b* allelic expression patterns, and then compared the results to predictions based on corresponding perturbations in the parallel *trans-cis* model.

We sorted DN2 cells with no expression, mono-allelic expression or bi-allelic expression of *Bcl11b* as initial populations, and then cultured them either on OP9-DL1 or OP9-Control feeders to maintain

or remove Notch signaling, respectively. After 4 days, we analyzed the resulting *Bcl11b* allelic expression states using flow cytometry (*Figure 5A*).

DL1 removal affected the distribution of *Bcl11b* allelic expression states differently depending on the state of the starting cell population. For progenitors with no initial *Bcl11b* expression, DL1 removal decreased the total fraction of cells that subsequently expressed *Bcl11b* from either allele from 0.9 to 0.5 (sum of mono-allelic and bi-allelic expressing cells, *Figure 5A*), as expected from the

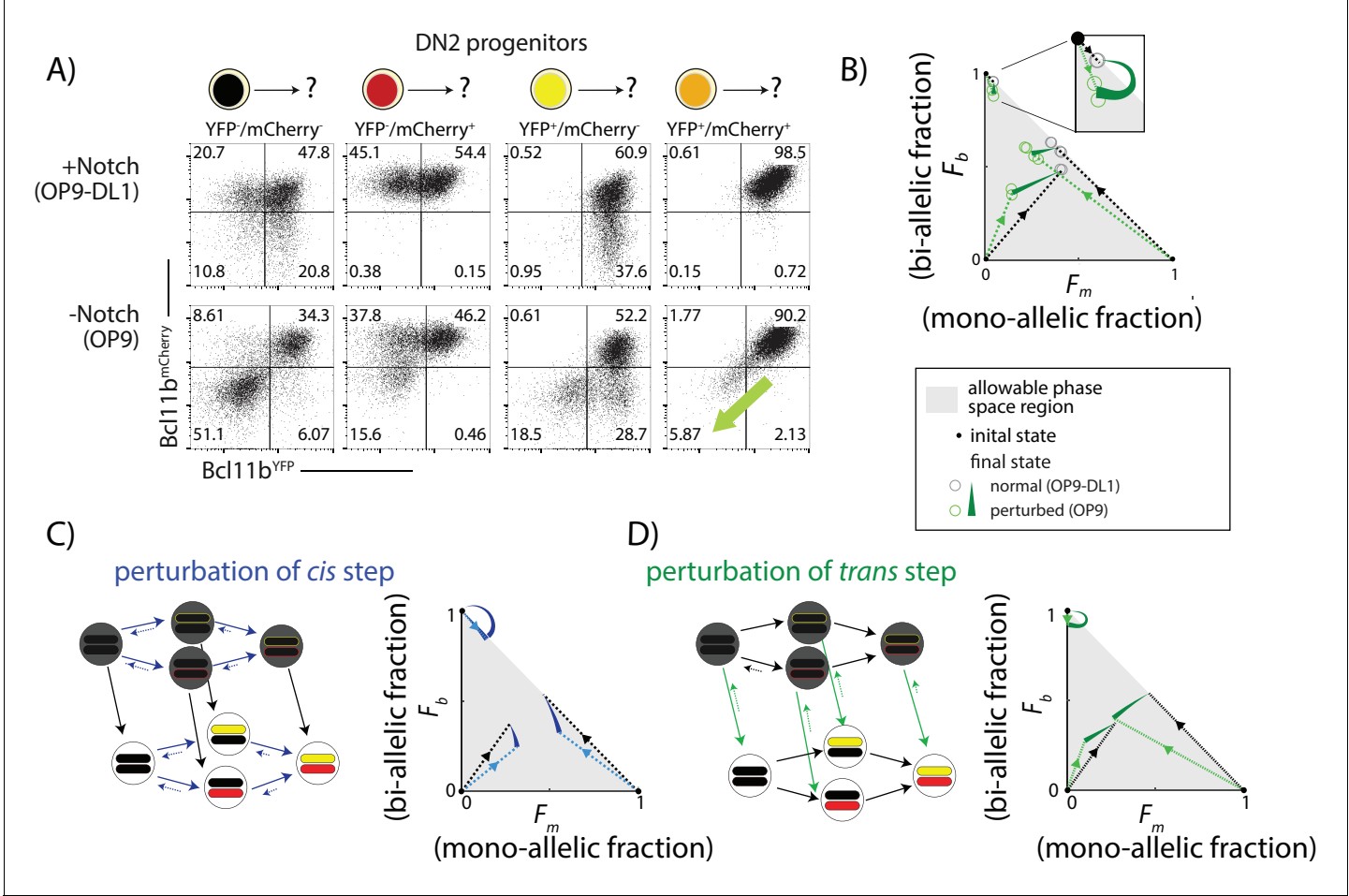

**Figure 5.** Notch signaling controls a parallel *trans*-acting step for *Bcl11b* activation. BM-derived DN2 progenitors with different *Bcl11b* allelic activation states were sorted, cultured on either OP9-Control (-Notch) or OP9-DL1 (+Notch) monolayers for four days, and analyzed using flow cytometry. (**A**) Flow cytometry plots show Bcl11b-mCherry versus Bcl11b-YFP expression levels in analyzed cells. Percentages of non-expressing, mono-allelic expressing (both YFP and mCherry) and bi-allelic expressing cells were used to calculate the locations in the phase space. Note that when Notch signaling is withdrawn from bi-allelically expressing cells, they downregulate both alleles coordinately (green-shaded arrow). (**B**) Phase space diagrams experimentally obtained from analysis of flow cytometry data. Points in phase space represent the average of 2–4 replicate data points in a single experiment (hollow circles). Inset shows final activation states of bi-allelic starting progenitors upon Notch withdrawal. Results shown are representative of two independent experiments. (**C–D**) Predicted phase space diagrams for fraction of bi-allelic expressing cells ($F_b$) against the fraction of mono-allelic expressing cells ($F_m$, YFP+ and mCh+ combined), for either the sequential *trans-cis* activation model (**C**), or the parallel *trans-cis* model (see Appendix for details). Black (colored) dotted lines connect initial state to the normal (perturbed) final state. Note that actual developmental trajectories may be curved (not shown). Arrows show predicted shifts in final state due to the indicated perturbations. Note that perturbations affect both the rates and reversibility of the indicated reactions. See also *Figure 5—figure supplement 1*.
DOI: https://doi.org/10.7554/eLife.37851.021

The following source data and figure supplement are available for figure 5:

**Source data 1.** Flow Cytometry Analysis of BM-derived DN2 progenitors cultured in the presence or absence of Notch.
DOI: https://doi.org/10.7554/eLife.37851.023

**Figure supplement 1.** Notch controls a parallel *trans*-acting step for *Bcl11b* activation.
DOI: https://doi.org/10.7554/eLife.37851.022

known positive effect of Notch on *Bcl11b* activation (*Kueh et al., 2016*). This reduction disproportionately affected mono-allelic expressing cells, such that the fraction of mono-allelic expressing cells relative to bi-allelic expressing cells decreased (from ~0.8 to 0.4, *Figure 5A*). In progenitors starting with mono-allelic *Bcl11b* expression, DL1 removal showed a large percentage reduction in the mono-allelic population, with a smaller reduction in the bi-allelic population (~14% versus 20% reduction, from *Figure 5A*). Finally, in bi-allelic expressing *Bcl11b* progenitors, most cells maintained expression despite Notch removal, as expected (*Kueh et al., 2016*), but a small fraction (~0.06) lost expression of both *Bcl11b* alleles entirely, reverting directly from the bi-allelic to a non-expressing state (*Figure 5A*, green arrows). To visually summarize these effects of Notch withdrawal, we represented the distribution of non-expressing, mono-allelic, and bi-allelic expressing cell states in the population in each condition from each starting population as points within a triangular region of allowed states in a single diagram (*Figure 5B*).

To interpret these experimental results, we compared the observed effects with predicted effects of a step-like perturbation in either the *cis*- or the *trans*-acting steps of the model (see Appendix). In order to account for reversibility in *Bcl11b* activation observed upon DL1 removal (*Figure 5A*), perturbation of the *cis* or *trans* step was implemented by both decreasing its forward rate and increasing its backwards rate. Additionally, all perturbations were assumed to have a weaker effect on transitions to or from the *Bcl11b* bi-allelic expressing state compared to transitions to or from the mono-allelic state. This assumption was designed to reflect the reduced impact of Notch signal withdrawal on cells starting with bi-allelic *Bcl11b* expression (*Bcl11b* inactivation was observed in ~6% of bi-allelic expressing progenitors versus ~15–18% of mono-allelic expressing progenitors, *Figure 5A*). We simulated the model with different strengths of *cis* or *trans* perturbations, and generated distributions of *Bcl11b* allelic activation states from non-expressing, mono-allelic, and bi-allelic starting populations.

Perturbation of the *cis*-acting step decreased the total fraction of cells expressing *Bcl11b* from all initial cell populations, as expected (*Figure 5C*). However, in contrast to experimental observations, this simulated perturbation increased, rather than decreased, the ratio of mono-allelic expressing cells to bi-allelic expressing cells (*Figure 5C*). It also caused bi-allelic expressing cells to sequentially turn off Bcl11b one allele at a time, rather than simultaneously as observed experimentally. Simulated perturbation of the *cis*-acting step thus did not match the observed effects of Notch withdrawal (*Figure 5B,C*).

By contrast, perturbation of the *trans*-acting step in the model produced effects resembling Notch withdrawal. First, it led to direct reversion of bi-allelic expressing progenitors to a non-expressing state, without passing through mono-allelic intermediates (*Figure 5D*, green arrows). Simultaneous inactivation of both alleles is difficult to reconcile with Notch affecting independent (*cis*) effects at each allele but is expected in response to removal of a *trans*-acting factor required for maintaining expression (*Figure 5B*). Second, unlike the *cis* perturbation, the *trans* perturbation did not increase the mono-allelic to bi-allelic ratio. In the simplest case, where all *trans* steps are uniformly affected by Notch, the *trans* perturbation is independent of the distribution of *cis* states (*Figure 5—figure supplement 1*), and therefore the mono- to bi-allelic ratio remains constant. When the *trans* perturbation more strongly affects mono-allelic cells, as we assume here, the mono- to bi-allelic ratio decreases, opposite to observed effects of the *cis* perturbation but consistent with experimental results.

Additionally, we also considered a third possibility in which Notch controls a necessary *trans*-acting step occurring strictly prior to *cis*-activation, as postulated by the sequential *trans-cis* model (*Figure 5—figure supplement 1*). In this case, progenitors that express one or both *Bcl11b* alleles would no longer be affected by Notch withdrawal, inconsistent with the experimental observations (*Figure 5B*). Taken together, these results strongly suggest that a separate Notch-dependent *trans*-acting event, occurring in parallel with Bcl11b locus activation, is necessary for Bcl11b activation and T-cell lineage commitment.

## Bcl11b activation can only occur over a limited developmental window

Given the finite rate of *cis*- and *trans*-activation steps, all cells would be expected to eventually activate both *Bcl11b* copies. However, a small fraction of cells were consistently found to express *Bcl11b* mono-allelically in thymic and peripheral T cell subsets (*Figures 1* and *3B,C* and *Figure 3—figure supplements 1,2*). This result suggested that activation might be possible only for a limited

time and that cells might lose competence to activate any still-silent *Bcl11b* locus as they develop. To test this hypothesis, we sorted mono-allelically expressing cells from different developmental stages, cultured them in vitro on OP9-DL1 monolayers for 4 days, and analyzed expression of both *Bcl11b* alleles (**Figure 6A**). The already-active copy retained active expression throughout the assay, as expected. However, the frequency of activation of the initial silent *Bcl11b* allele varied strongly with developmental stage. Activation occurred efficiently at the DN2 stage (DN2A and DN2B combined) but dropped sharply as cells progressed to DN3 (~80% versus~15% activated after 4 days, **Figure 6A**), and dropped even further at the double positive (DP) and CD4 single positive stages (~1.5% and 2.4%, respectively, **Figure 6A**). Equivalent results were obtained regardless of whether

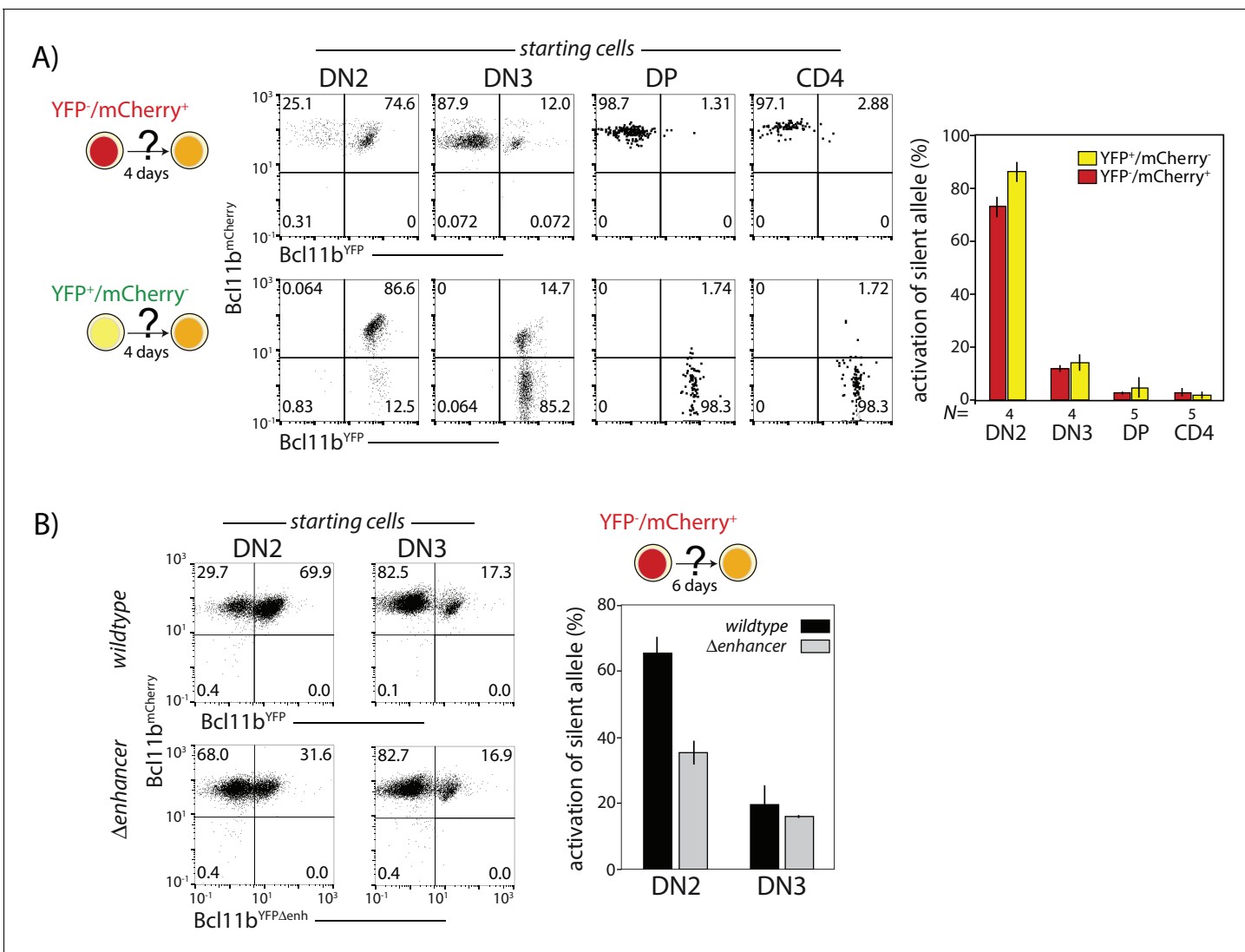

**Figure 6.** Probabilistic *Bcl11b* activation occurs within a limited developmental time window. Cells expressing only one *Bcl11b* allele at the indicated stages were sorted from thymocytes, cultured for 4d on OP9-DL1 monolayers, and analyzed for activation of the initially inactive *Bcl11b* allele using flow cytometry. (A) Flow plots (left) show Bcl11b-mCh versus Bcl11b-YFP expression levels for descendants of cells that had mono-allelic expression at the indicated stages of development; bar charts (right) show the fraction of progenitors from different stages that activate the silent *Bcl11b* allele upon culture. Data represent mean and standard deviation of 4–5 replicates, derived from two independent experiments. The competence to activate the silent *Bcl11b* allele decreases upon progression to the DN3 stage and beyond. (B) Flow plots (left) show Bcl11b-mCh versus Bcl11b-YFP expression levels for DN2 or DN3 progenitors with either an intact YFP allele enhancer (top) or a disrupted YFP allele enhancer (bottom). Bar chart (right) shows the fraction of cells activating the silent *Bcl11b* allele upon re-culture. Data show that enhancer disruption reduces the *Bcl11b* activation advantage in DN2 cells as compared to DN3 cells. Data represent mean and standard deviation of three replicates from two independent experiments.
DOI: https://doi.org/10.7554/eLife.37851.024

the experiment started with active wildtype YFP and mCherry alleles (*Figure 6A*). These results indicate that *cis*-activation of *Bcl11b* predominantly occurs during DN2 and DN3 stages.

This DN2-stage preference for *Bcl11b* activation competence could arise from stage-specific activity of the identified distal enhancer. To test this hypothesis, we compared the activation kinetics of intact and enhancer-disrupted YFP alleles in sorted progenitors expressing only the *Bcl11b* mCherry allele. When the input cells were DN2 cells, the enhancer-disrupted YFP allele showed markedly less activation over the next four days than the intact YFP allele (70% versus 32%, *Figure 6B*). However, using input cells sorted at the DN3 stage, no differences in activation propensity were observed, with both wildtype and disrupted enhancer alleles showing the same attenuated degree of activation (~17%). These results suggest that the *Bcl11b* enhancer works specifically to enhance *cis*-activation of *Bcl11b* at the DN2 stage.

## Discussion

Stochastic epigenetic control switches have been described in yeasts, plants, and, more recently, constructed in synthetic systems (*Berry et al., 2017*; *Bintu et al., 2016*; *Hathaway et al., 2012*; *Keung et al., 2014*; *Xu et al., 2006*), yet their roles in controlling fate decisions in vertebrate developmental systems are not well understood. Specifically, it is not clear when epigenetic states simply respond passively to 'upstream' developmental changes in transcription factor activity, and when they actively impose distinct temporal constraints on transcription factor effects. By separately following the two chromosomal copies of *Bcl11b* in single cells, we found that the decision to turn on *Bcl11b*, and the ensuing transition to T-cell fate, involves a stochastic, irreversible rate-limiting *cis*-activation step that occurs on each chromosomal allele of the *Bcl11b* gene itself. The *cis*-acting step occurs at a low enough rate ($k_C = (4.2 \pm 3.3) \times 10^{-3}$/hr, *Figure 4A*) to generate numerous mono-allelically expressing cells as intermediates, and is stable enough to propagate the same mono-allelic activation state through multiple rounds of cell division in individual clones. In particular, by generating delays of multiple days and cell generations prior to differentiation, the *cis*-acting switch also indirectly controls the overall degree of proliferation of the progenitor pool. These results thus demonstrate that stochastic, epigenetic events on individual gene loci can fundamentally limit the timing and outcome of mammalian cell fate decisions, as well as the population structure of the resulting differentiated population.

Slow, stochastic *Bcl11b* activation is controlled in part by an enhancer far downstream from the *Bcl11b* promoter, on the opposite end of the same topologically associated domain. Multiple known epigenetic changes that occur on the *Bcl11b* locus could participate in the processes whose dynamics we have measured here. The distal enhancer could recruit chromatin regulators that clear repressive chromatin modifications from the *Bcl11b* locus. In its silent state, the *Bcl11b* promoter and gene body are covered by DNA methylation and histone H3K27me3 modifications (*Hu et al., 2018*; *Ji et al., 2010*; *Zhang et al., 2012*). Chromatin regulators recruited by the enhancer could disrupt repressive modifications in their vicinity, catalyzing a phase transition that results in cooperative, all-or-none removal of repressive marks on the entire gene locus (*Larson et al., 2017*; *Strom et al., 2017*). As another possibility, the distal enhancer could recruit *trans*- factors that facilitate its T-lineage-specific looping with the *Bcl11b* promoter and its subsequent activation (*Li et al., 2013*). In early T-cell progenitors, the *Bcl11b* promoter establishes new contacts with its distal enhancer, resulting in de novo formation of an altered topological associated domain, with boundaries defined by these two elements (*Hu et al., 2018*; *Isoda et al., 2017*). *Trans*- regulators of DNA loop extrusion that associate with the distal enhancer, whose binding may be facilitated by non long-coding RNA transcription (*Isoda et al., 2017*), may stabilize these looping interactions (*Fudenberg et al., 2016*; *Nasmyth, 2001*; *Riggs, 1990*; *Sanborn et al., 2015*), which may release *Bcl11b* from the repressive environment of the nuclear periphery and permit its activation (*Isoda et al., 2017*). The evidence for such epigenetic differences associated with the *Bcl11b* locus in T and non-T cells have been known for some time, but the functional impacts of *cis*-acting mechanisms on locus activation dynamics has been unknown until now. Ultimately, any of these mechanisms that are rate-limiting will have to account for the stochastic nature of *Bcl11b* locus activation, its exceptionally long activation time constant, and its all-or-none, irreversible nature, demonstrated here. Dissecting the molecular and biophysical basis of these striking emergent properties will be the subject of future investigation.

Mathematical modeling, together with perturbation analysis, are consistent with *Bcl11b* expression requiring a separate Notch signal-dependent *trans-* event that is needed in parallel with *Bcl11b* *cis-*activation (*Figure 7*). The comparable slow rate constants for parallel *cis* and *trans* steps imply that a substantial fraction of cells can undergo the *cis-*acting step prior to *trans-*activation and observable *Bcl11b* expression. Furthermore, although the experiments here all start with DN2A progenitors, relevant dynamics may extend to earlier stages. In fact, the *cis-*acting step could potentially occur within the preceding ETP stage or during the ETP-DN2a transition (*Kueh et al., 2016*). Consistent with this hypothesis, changes in *Bcl11b* chromatin state associated with gene activation can already be observed at the ETP stage (*Isoda et al., 2017*; *Zhang et al., 2012*). Furthermore, previous work showed that knockdown of Gata3 and TCF-1, which first turn on at the ETP stage and are required for Bcl11b activation but not for subsequent events, impact Bcl11b activation more strongly in ETP cells than in DN2A cells that have not yet activated Bcl11b, suggesting that many cells may enter DN2A after *cis-*activation has already occurred. Looking ahead, a more complete model will therefore have to span multiple stages of T cell development.

How widespread are stochastic epigenetic switches of the type analyzed here? The only other regulatory switch whose individual allele dynamics been similarly characterized, to our knowledge, occurs in the plant vernalization system, which controls flowering in response to periods of cold temperature. Specifically, exposure to cold causes silencing of *FLC*, a master repressor of flowering. An obvious regulatory difference between the two systems is in the direction of regulation, with *FLC* undergoing silencing and *Bcl11b* undergoing activation. Nevertheless, the two systems share common dynamic features. Like *Bcl11b*, *FLC* silencing involves an all-or-none switch that occurs stochastically, independently at distinct gene copies in the same cell, and is stably inherited during cell division (*Angel et al., 2011*; *Berry et al., 2015*; *Yang et al., 2017*). Also like *Bcl11b*, silencing of the FLC locus occurs at rates lower than that of cell division, giving rise to distinct mono- and bi-allelic expressing states that persist over multiple cell generations. In the vernalization system, *FLC* silencing was shown to occur in two steps: First, repressive H3K27me3 modifications nucleate near the promoter, producing a metastable (reversible) silent state. Second, these marks spread across the locus, locking the locus into a more stable silent state (*Yang et al., 2017*). With *Bcl11b*, we do

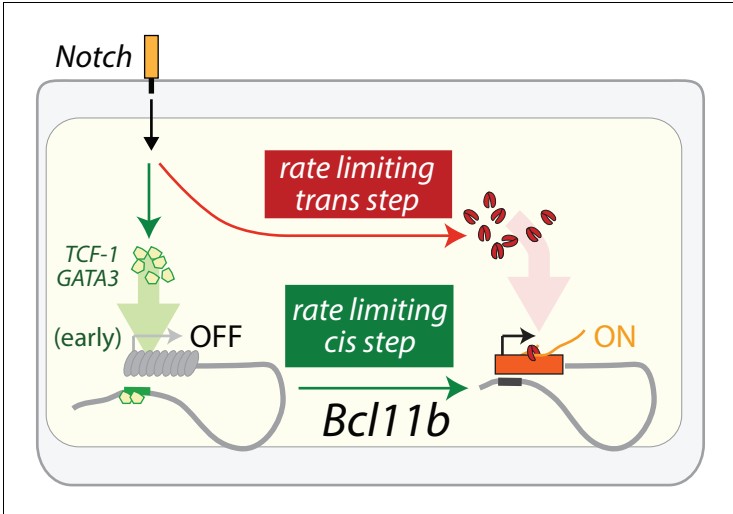

**Figure 7.** Model of *Bcl11b* regulation by parallel *cis* and *trans*-limiting steps. *Bcl11b* activation requires two rate-limiting steps: a switch of the *Bcl11b* locus from an inactive to active epigenetic state, and the activation of a *trans* factor is necessary for transcription of *Bcl11b* from an activated locus. Notch signaling activates TCF-1 and GATA3 in early thymic progenitors (*García-Ojeda et al., 2013*; *Scripture-Adams et al., 2014*; *Weber et al., 2011*), and these two factors may act on the identified distal enhancer to control the rate-limiting *cis* step on the *Bcl11b* locus (green). In parallel, Notch promotes the activation of a *trans* factor (red) that is necessary for transcription from a *cis*-activated *Bcl11b* locus. The *cis* and *trans*-limiting steps together control the dynamics of Bcl11b expression and T-cell lineage commitment.
DOI: https://doi.org/10.7554/eLife.37851.026

not yet know whether chromatin modifications are causally responsible for activation, nor do we know whether the gene passes through a metastable intermediate as *FLC* does. In both systems, upstream *trans*-acting factors – Vin3 for *FLC* and Gata-3 and TCF-1 for *Bcl11b* – control the stochastic rate of silencing or activation, respectively, but do not deterministically specify the transcription rates of the individual gene. For *FLC*, it is not yet known whether additional parallel trans-acting steps are also required, as is the case with *Bcl11b*. Analyzing the dynamics of allelic silencing using the framework described above could provide insight into this question. While the schemes likely differ between the two systems, the many similarities in the dynamics of regulation between these two very different contexts suggest that stochastic epigenetic switches are likely to be prevalent.

Slow, stochastic epigenetic switches, similar to the one we describe here, may allow cells to tune the size and composition of differentiated tissues. By using *trans*-acting inputs that modulate activation probabilities, such epigenetic switches could translate differences in input duration to changes in the fraction of output cells activated (*Bintu et al., 2016*), a strategy that could enable tunable control of cellular proportions in a developing tissue or organ. Moreover, a striking aspect of this mechanism is its ability to generate populations of mature T cells that are mosaic in the status of their activation of the two *Bcl11b* alleles. Indeed, the differential distribution of mono-allelically expressing cells that we see among distinct functional T-cell subsets suggests the potential of non-uniform allelic activity to alter function or selective fitness. The increased fraction of mono-allelically expressing cells that appear when an enhancer complex is weakened is a strong phenotype at the single cell level that could be relevant to enhancer polymorphisms in natural populations, although its impact could easily be underestimated by more conventional gene expression analyses.

Here, we have illustrated a general approach that can reveal the dynamics of epigenetic control mechanisms, determine their prevalence in the genome, and elucidate their functional roles in multicellular organism development and function. Stochastic epigenetic switches, similar to the one uncovered here, may constitute fundamental building blocks of cell fate control circuits in mammalian cells. As cells transition from one developmental state to another, they undergo concerted transformations in the chemical modification states or physical conformations of many regulated genes. These changes could reflect more widespread roles for epigenetic mechanisms in controlling cell state transition timing.

# Materials and methods

**Key resources table**

| Reagent type (species) or resource | Designation | Source or reference | Identifiers | Additional information |
|---|---|---|---|---|
| Recombinant DNA reagent | pTarget Bcl11b IRES-H2BmCherry-neo/3pUTR | This paper | N/A | Gene targeting vector with IRES-H2B-mCherry-loxP-neo-loxP cassette knocked into 3' UTR of Bcl11b |
| Recombinant DNA reagent | pTarget Bcl11b dEnh-hygro | This paper | N/A | Gene targeting vector with Enhancer replaced by hygromycin cassette |
| Recombinant DNA reagent | FRT-PGK-gb2-hygromycin-FRT cassette | Genebridges | Cat# A010 | |
| Recombinant DNA reagent | MSCV IRES H2B-mCerulean | *Kueh et al., 2013* | N/A | |
| Recombinant DNA reagent | pCL-Eco | Imgenex | Cat# NBP2-29540 | |

*Continued on next page*

*Continued*

| Reagent type (species) or resource | Designation | Source or reference | Identifiers | Additional information |
|---|---|---|---|---|
| Strain, strain background (mouse) | Bcl11b<sup>YFP(neo)/mCh(neo)</sup> | This paper | N/A | Two color reporter mice generated from breeding animals homozygous for either Bcl11b YFP(neo) or Bcl11b mCh(neo). See Materials and methods for details. |
| Strain, strain background (mouse) | Bcl11b<sup>mCh(neo)/mCh(neo)</sup> | This paper | N/A | Homozygous Bcl11b mCh(neo) reporter mice used to generate two color reporter mice. Derived from Bcl11b YFP/mCh(neo) F0 chimeric mice. See Materials and methods for details. |
| Strain, strain background (mouse) | Bcl11b<sup>YFP/mCh(neo)</sup> | This paper | N/A | Control mice for comparing the effects of the enhancer on Bcl11b expression. Generated by targeting Bcl11b mCherry gene targeting vector to V6.5 mouse embryonic stem (ES) cells with single modified Bcl11b mCitrine dneo allele. See Materials and methods for details. |
| Strain, strain background (mouse) | Bcl11b<sup>YFPdEnh/mCh(neo)</sup> | This paper | N/A | Two color reporter mouse with Bcl11b enhancer deleted. Generated by targeting dEnh gene target vector to V6.5 mouse ES cells with genotype Bcl11b YFP/mCh(neo). See Materials and methods for details. |
| Strain, strain background (mouse) | Bcl11b<sup>YFPdEnh/dEnh</sup> | This paper | N/A | Homozygous deleted enhancer mice generated from Bcl11b YFP dEnh/mCh(neo) mice. See Materials and methods for details. |
| Strain, strain background (mouse) | CD45.1 C57BL/6: B6.SJL-*Ptprc*<sup>a</sup> *Pepc*<sup>b</sup>/BoyJ | Jackson Laboratory | Stock No# 002014 | |
| Cell line (mouse) | OP9-DL1-GFP | *Schmitt and Zúñiga-Pflücker, 2002* | N/A | |
| Cell line (mouse) | OP9-Mig | *Schmitt and Zúñiga-Pflücker, 2002* | N/A | |
| Cell line (mouse) | OP9-DL1-hCD8 | *Kueh et al., 2016* | N/A | |
| Cell line (human) | Human Phoenix-ECO | ATCC | Cat# CRL-3214 | |
| Antibody | Anti-mouse CD8a Biotin (clone 53–6.7) | eBioscience | Cat# 13-0081-86; RRID:AB_466348 | (1:100) |
| Antibody | Anti-mouse TCRb Biotin (clone H57-597) | eBioscience | Cat# 13-5961-85; RRID:AB_466820 | (1:100) |
| Antibody | Anti-mouse TCRgd Biotin (clone GL3) | eBioscience | Cat# 13-5711-85; RRID:AB_466669 | (1:100) |
| Antibody | Anti-mouse Ter119 Biotin (clone TER-119) | eBioscience | Cat# 13-5921-85; RRID:AB_466798 | (1:100) |
| Antibody | Anti-mouse NK1.1 Biotin (clone PK136) | eBioscience | Cat# 13-5941-85; RRID:AB_466805 | (1:100) |

*Continued on next page*

*Continued*

| Reagent type (species) or resource | Designation | Source or reference | Identifiers | Additional information |
|---|---|---|---|---|
| Antibody | Anti-mouse Gr-1 Biotin (clone RB6-8C5) | eBioscience | Cat# 13-5931-86; RRID:AB_466802 | (1:100) |
| Antibody | Anti-mouse CD11c Biotin (clone N418) | eBioscience | Cat# 13-0114-85; RRID:AB_466364 | (1:100) |
| Antibody | Anti-mouse CD11b Biotin (clone M1/70) | eBioscience | Cat# 13-0112-86; RRID:AB_466361 | (1:100) |
| Antibody | Anti-mouse CD19 Biotin (clone 1D3/6D5) | eBioscience | Cat# 13-0193-85; RRID:AB_657658 | (1:100) |
| Antibody | Anti-mouse CD3e Biotin (clone 145–2 C11) | eBioscience | Cat# 13-0031-85; RRID:AB_466320 | (1:100) |
| Antibody | Anti-human/mouse B220 Biotin (clone RA3-6B2) | eBioscience | Cat# 13-0452-85; RRID:AB_466450 | (1:100) |
| Antibody | Anti-mouse F4/80 Biotin (clone BM8) | eBioscience | Cat# 13-4801-85; RRID:AB_466658 | (1:100) |
| Antibody | Anti-mouse CD4 Biotin (clone GK1.5) | eBioscience | Cat# 13-0041-85; RRID:AB_466326 | (1:100) |
| Antibody | Anti-human/mouse CD44 eFluor 450 (clone IM7) | eBioscience | Cat# 48-0441-82; RRID:AB_1272246 | (1:300) |
| Antibody | Anti-mouse CD25 Brilliant Violet 510 (clone PC61) | Biolegend | Cat# 102041; RRID:AB_2562269 | (1:300) |
| Antibody | Anti-mouse CD117 (cKit) APC-eFluor 780 (clone 2B8) | eBioscience | Cat# 47-1171-82; RRID:AB_1272177 | (1:300) |
| Antibody | Anti-mouse HSA eFluor 450 (clone M1/69) | eBioscience | Cat# 48-0242-82; RRID:AB_1311169 | (1:300) |
| Antibody | Anti-mouse CD4 Brilliant Violet 510 (clone GK1.5) | Biolegend | Cat# 100449; RRID:AB_2564587 | (1:300) |
| Antibody | Anti-mouse CD8a APC (clone 53–6.7) | eBioscience | Cat# 17-0081-82; RRID:AB_469335 | (1:300) |
| Antibody | Anti-mouse TCRb APC-eFluor 780 (clone H57-597) | eBioscience | Cat# 47-5961-82; RRID:AB_1272173 | (1:300) |
| Antibody | Anti-mouse CD25 APC-eFluor 780 (clone PC61.5) | eBioscience | Cat# 47-0251-82; RRID:AB_1272179 | (1:300) |
| Antibody | Anti-mouse CD19 eFluor 450 (clone 1D3/6D5) | eBioscience | Cat# 48-0193-82; RRID:AB_2734905 | (1:300) |
| Antibody | Anti-mouse CD117 (cKit) APC (clone 2B8) | eBioscience | Cat# 17-1171-82; RRID:AB_469430 | (1:300) |
| Antibody | Anti-mouse CD45 APC-eFluor 780 (clone 30-F11) | eBioscience | Cat# 47-0451-82; RRID:AB_1548781 | (1:300) |
| Antibody | Anti-mouse CD25 APC (clone PC61.5) | eBioscience | Cat# 17-0251-82; RRID:AB_469366 | (1:300) |
| Antibody | Anti-mouse CD4 APC-eFluor 780 (clone GK1.5) | eBioscience | Cat# 47-0041-82; RRID:AB_11218896 | (1:300) |

*Continued on next page*

*Continued*

| Reagent type (species) or resource | Designation | Source or reference | Identifiers | Additional information |
|---|---|---|---|---|
| Antibody | Anti-mouse CD8a APC-eFluor 780 (clone 53–6.7) | eBioscience | Cat# 47-0081-82; RRID:AB_1272185 | (1:300) |
| Antibody | Anti-mouse CD45 APC (clone 30-F11) | eBioscience | Cat# 17-0451-82; RRID:AB_469392 | (1:300) |
| Antibody | Anti-mouse CD5 eFluor 450 (clone 53–7.3) | eBioscience | Cat# 48-0051-82; RRID:AB_1603250 | (1:300) |
| Antibody | Anti-mouse TCRgd APC (clone GL3) | eBioscience | Cat# 17-5711-82; RRID:AB_842756 | (1:300) |
| Antibody | Anti-mouse CD49b eFluor 450 (clone DX5) | eBioscience | Cat# 48-5971-82; RRID:AB_10671541 | (1:300) |
| Antibody | Anti-mouse NK1.1 APC (clone PK136) | eBioscience | Cat# 17-5941-82; RRID:AB_469479 | (1:300) |
| Antibody | Anti-mouse CD3e APC-eFluor 780 (clone 145–2 C11) | eBioscience | Cat# 47-0031-82; RRID:AB_11149861 | (1:300) |
| Antibody | Anti-mouse TCRb eFluor 450 (clone H57-597) | eBioscience | Cat# 48-5961-82; RRID:AB_11039532 | (1:300) |
| Antibody | Anti-mouse CD49b Biotin (clone DX5) | eBioscience | Cat# 13-5971-82; RRID:AB_466825 | (1:300) |
| Antibody | Anti-mouse CD62L APC (clone MEL-14) | eBioscience | Cat# 17-0621-82; RRID:AB_469410 | (1:300) |
| Antibody | Anti-mouse CD45.2 Brilliant Violet 510 (clone 104) | Biolegend | Cat# 109837; RRID:AB_2561393 | (1:300) |
| Antibody | Anti-mouse CD4 eFluor 450 (clone GK1.5) | eBioscience | Cat# 48-0041-82; RRID:AB_10718983 | (1:300) |
| Antibody | Anti-mouse CD45 eFluor 450 (clone 30-F11) | eBioscience | Cat# 48-0451-82; RRID:AB_1518806 | (1:300) |
| Antibody | Streptavidin PerCP-Cyanine5.5 | Biolegend | Cat# 405214; RRID:AB_2716577 | (1:300) |
| Antibody | Streptavidin Brilliant Violet 510 | Biolegend | Cat# 405234 | (1:300) |
| Peptide, recombinant protein | Recombinant Human Flt3-Ligand | PeproTech | Cat# 300–19 | |
| Peptide, recombinant protein | Recombinant Human IL-7 | PeproTech | Cat# 200–07 | |
| Peptide, recombinant protein | Recombinant Human Stem Cell Factor (SCF) | PeproTech | Cat# 300–07 | |
| Peptide, recombinant protein | Recombinant Mouse IL-6 | eBioscience | Cat# 14-8061-62 | |
| Peptide, recombinant protein | Recombinant Mouse Stem Cell Factor (SCF) | eBioscience | Cat# 34-8341-82 | |
| Peptide, recombinant protein | Recombinant Mouse IL-3 | eBioscience | Cat# 14-8031-62 | |
| Peptide, recombinant protein | Retronectin | Takara | Cat# T100B | |
| Peptide, recombinant protein | DL1-extIgG Protein | *Varnum-Finney et al., 2000* | N/A | |

*Continued on next page*

*Continued*

| Reagent type (species) or resource | Designation | Source or reference | Identifiers | Additional information |
|---|---|---|---|---|
| Software, algorithm | FlowJo (v10.0.8) | Tree Star | N/A | |
| Software, algorithm | MATLAB (R2016a) | MathWorks | N/A | |
| Other | FuGENE 6 Transfection Reagent | Promega | Cat# E2691 | |
| Other | MACS Streptavidin Microbeads | Miltenyi Biotec | Cat# 130-048-101 | |
| Other | LS Columns | Miltenyi Biotec | Cat# 130-042-401 | |
| Other | 250mm-diameter PDMS circular micromesh arrays | Microsurfaces Pty Ltd | Cat# MMA-0250-100-08-01 | |

## Experimental model and subject details

### Animals

$F_o$ chimeric mice from Bcl11b$^{YFP/mCh(neo)}$ and Bcl11b$^{YFP\Delta Enh/mCh(neo)}$ ES-cell blastocyst injections were all made in our lab (described in Materials and method Details). Founder animals were brought to term and crossed in house to generate Bcl11b$^{YFP(neo)/mCh(neo)}$, Bcl11b$^{YFP/mCh(neo)}$, Bcl11b$^{YFP\Delta Enh/mCh(neo)}$, and Bcl11b$^{YFP\Delta Enh/mCh(neo)}$ mice. CD45.1 C57BL/6 mice were purchased from Jackson Laboratory. All adult animals were used between 5 and 12 weeks of age. Both male and female mice were used similarly in all studies. Animals used for these experiments were bred and maintained at the Animal Facilities at both the California Institute of Technology and the University of Washington, and animal protocols were reviewed and approved by the Institute Animal Care and Use Committees of both institutions (Protocols #1445 and #1409, California Institute of Technology; Protocol #4397–01, University of Washington).

### Cells

Primary cells isolated from thymus, spleen, bone marrow, and fetal livers were cultured on a OP9-DL1 or OP9-control stromal monolayer system (*Schmitt and Zúñiga-Pflücker, 2002*) at 37°C in 5% $CO_2$ conditions with standard culture medium [80% αMEM (Gibco), 20% Fetal Bovine Serum (Sigma-Aldrich), Pen-Strep-Glutamine (Gibco), 50 μM β-mercaptoethanol (Sigma)] supplemented with appropriate cytokines (described in Materials and method Details). Both OP9-DL1 and OP9-control cell lines were tested and found to be negative for mycoplasma contamination.

## Method details

### Construct designs

Gene targeting vectors for generating dual allelic *Bcl11b* fluorescent reporter and subsequent enhancer knockout were constructed using a two-step bacterial artificial chromosome (BAC) recombineering method. First, *Bcl11b*-BACs were modified to either insert a fluorescent reporter or disrupt the enhancer sequence with a drug selection marker. An internal ribosome entry site (IRES)-histone 2B-mCherry red fluorescent protein (mCh)-*loxP*-neomycin (*neo*)-*loxP* cassette with homology arms targeting the 3'-untranslated region (UTR) of *Bcl11b* was derived from a similar histone 2B-mCitrine yellow fluorescent protein (YFP) gene targeting vector version published previously (*Kueh et al., 2016*) and an IRES-H2B-mCherry-*loxP*-neomycin (*neo*)-*loxP* cassette. These two starting plasmids were digested with restriction enzymes NheI and HindIII (New England Biolabs) to exchange the fluorescent protein sequences. Homology arms flanking the 5' and 3' ends of the 1.9 kb enhancer (Enh) sequence to be replaced (chr12: 108,396,825–108,398,612, mm9 assembly; chr12:107,158,615–107,160,462, in mm10) were attached to a *FRT*-PGK-gb2-hygromycin (*hygro*)-*FRT* drug selection cassette through fusion PCR, and inserted into a cloning vector (pGEM-T-Easy, Promega). Next, restriction enzymes were used to release the homology-flanked fluorescent or drug

reporter cassettes, and the resultant linear fragments were introduced into recombineering *E. Coli* strain SW102 containing appropriate BACs for specific targeting. The IRES-mCh-*neo* fragment was linearized with AatII, SalI-HF, ScaI-HF and knocked into a BAC containing the entire *Bcl11b* gene locus (RP24-282D6, from http://bacpac.chori.org). Restriction enyzmes XmnI, PspOMi, and SbfI released the *FRT-hygro-FRT* cassette used to replace the enhancer sequence in a BAC containing genomic regions downstream of the *Bcl11b* locus (RP23-445J15, from http://bacpac.chori.org). Correctly modified BACs were then selected using kanamycin or hygromycin in combination with chloramphenicol, and verified by PCR and pulse-field gel electrophoresis analysis using the restriction enzyme NotI (New England Biolabs).

A second recombineering reaction retrieved the targeting sequences from reporter modified *Bcl11b*-BACs. The retrieval vector used to fetch the targeting sequence from the modified *Bcl11b*-mCherry-*neo* BAC was made in a previous study (*Kueh et al., 2016*). For retrieval of the enhancer-disrupted sequence, homology arms for retrieval were first generated using fusion PCR, then cloned into a vector containing a Herpes Simplex Virus-Thymidine Kinase (HSV-TK) cassette using restriction enzymes NotI and SpeI (New England Biolabs). Both retrieval vectors were linearized with PacI and AscI (New England Biolabs), introduced into SW102 containing respective modified Bcl11b-BACs, and retrieved targeting sequences between the homologous ends to generate the desired gene targeting vectors. Clones that underwent correct retrieval reactions were selected using kanamycin or hygromycin in combination with ampicillin, and verified with restriction enzyme digests and sequencing.

The retroviral construct expressing IRES-H2B-mCerulean cyan fluorescent protein (CFP) used for timelapse imaging experiments was generated in a previous study (*Kueh et al., 2013*). A complete list of vectors used is provided in Key Resources Table.

## Mouse generation

A series of genetic modifications were performed to generate different *Bcl11b* reporter mouse strains used for this study (*Figure 1—figure supplement 1*). V6.5 mouse embryonic stem (ES) cells with a single modified *Bcl11b* allele expressing the IRES-H2B-mCitrine-*loxp-neo-loxp* fluorescent reporter were first transfected with Cre recombinase to excise the neomycin cassette. Subclones of this line with a correct deletion of the neomycin cassette were then targeted with the IRES-mCherry-*neo* gene targeting vector to generate dual allelic Bcl11b fluorescent reporter cells, and targeted again with the ΔEnh-*hygro* cassette to delete the enhancer in one allele. After each targeting event, recombinant ES cells grown on feeders were positively selected with antibiotics according to the cassette inserted, and negatively selected with G418. Resistant clones were passaged onto feeder-free conditions and screened using PCR and qPCR for correct targeting. Clones with the desired genotype were karyotyped for normal chromosome numbers before being injected into C57BL/6 blastocyst embryos or subjected to subsequent gene targeting.

$F_0$ chimeric mice from $Bcl11b^{YFP/mCh(neo)}$ and $Bcl11b^{YFPΔEnh/mCh(neo)}$ ES-cell blastocyst injections were generated, and either analyzed at embryonic day 14.5 (E14.5) or brought to term for breeding. $Bcl11b^{YFP/mCh(neo)}$ $F_0$ chimeric mice were crossed to C57BL/6 mice, and the offspring containing *Bcl11b*-IRES-mCherry-*neo* allele were then bred to homozygosity for this allele. Dual allelic $Bcl11b^{YFP(neo)/mCh(neo)}$ mice with identical *Bcl11b* alleles except for fluorescent protein reporters were generated from breeding $Bcl11b^{mCh(neo)/mCh(neo)}$ mice to previously produced $Bcl11b^{YFP(neo)/YFP(neo)}$ mice (*Kueh et al., 2016*), and were used for in vitro assay studies of bone marrow derived T-cells. $Bcl11b^{YFPΔEnh/mCh(neo)}$ mice were generated in a similar manner by first breeding to C57BL/6 mice to generate enhancer deleted heterozygotes, then crossing mice to $Bcl11b^{mCh(neo)/mCh(neo)}$. $Bcl11b^{YFPΔEnh/YFPΔEnh}$ mice were generated in parallel by crossing enhancer deleted heterozygotes together. For experiments comparing the effects of the enhancer on *Bcl11b* expression, direct control $Bcl11b^{YFP/mCh(neo)}$ mice were generated from breeding $Bcl11b^{YFP/YFP}$ and $Bcl11b^{mCh(neo)/mCh(neo)}$ animals. However, we have previously reported that the presence or absence of *neo* cassette does not affect the *Bcl11b* reporter locus (*Kueh et al., 2016*), and do not observe any differences in expression pattern in this study as well (see *Figures 1* and *3*).

## Cell purification

Thymocytes and splenocytes were purified from lymphoid organs removed from 4- to 6-week-old normal and enhancer-deleted two-color *Bcl11b* reporter strains, and 2-month post-fetal liver precursor transplantation CD45.1 chimeras prior to flow cytometry analysis or fluorescent activated cell sorting (FACS). Harvested lymphoid organs were mechanically dissociated to make single cell suspensions that were re-suspended in Fc blocking solution with 2.4G2 hybridoma supernatant (prepared in the Rothenberg lab). Early stage thymocyte precursors to be analyzed (ETP, DN2A, DN2B, DN3: *Figures 1C* and *3B*, and *Figure 3—figure supplement 1*) or sorted (DN2, DN3: *Figures 3D* and *6A*), were first depleted of mature cell lineages using a biotin-streptavidin-magnetic bead removal method. Thymocyte suspensions were labeled with biotinylated lineage marker antibodies (CD8α, TCRβ, TCRγδ, Ter119, Gr-1, CD11c, CD11b, NK1.1), incubated with MACS Streptavidin Microbeads (Miltenyi, Biotec) in HBH buffer (HBSS (Gibco), 0.5% BSA (Sigma-Aldrich), 10 mM HEPES, (Gibco)) pre-filtered through cell separation magnet (BD Biosciences), and passed through a magnetic column (Miltenyi Biotec). Rare T-cell subsets found in the spleen (*Figure 3—figure supplement 2*) were enriched using a similar depletion protocol by labeling splenocytes with biotinylated antibodies CD19, CD11b, CD11c, and Gr-1. Later-stage thymocyte precursors analyzed (*Figure 1D*, *Figure 3B*, and *Figure 3—figure supplements 1*, *3*, *4*) or sorted (*Figure 6A*), and whole splenocyte populations analyzed (*Figure 3—figure supplements 2*, *3*) were directly stained with conjugated fluorescent cell surface antibodies (see *Supplementary file 1*, Key Resources Table).

Bone Marrow (BM) cells were harvested from dissected femurs and tibiae of 2- to 3- month-old $Bcl11b^{YFP(neo)/mCh(neo)}$ mice. Fetal livers (FLs) were removed from $F_0$ chimeric fetuses of pregnant surrogate mice at E14.5, individually disrupted mechanically via pipetting into whole organ suspension, and frozen down in freezing media (50% FBS, 40% αMEM, 10% DMSO) for liquid nitrogen storage. Prior to in vitro culture use, BM and thawed FL cell suspensions were blocked in 2.4G2 supernatant, tagged with biotinylated antibody lineage markers specific to BM (CD19, CD11b, CD11c, NK1.1, Ter119, CD3ε, Gr-1, B220) or FL (CD19, F4/80, CD11c, NK1.1, Ter119, Gr-1), and depleted of biotin-streptavidin-magnetically labeled mature lineage cells as described above. Eluted lineage depleted (Lin⁻) bone marrow progenitors were either frozen down in freezing media for storage in liquid nitrogen or used directly for in vitro cell culture assays of T-cell development, while Lin⁻ fetal liver progenitors were immediately cultured.

## In vitro differentiation of T-cell progenitors

DN T-cell precursors used for in vitro studies were generated by culturing BM and FL stem and progenitor cells on a OP9-DL1 stromal monolayer culture system (*Schmitt and Zúñiga-Pflücker, 2002*), following previously detailed methods (*Kueh et al., 2016*) with adapted variations as described below. To promote the DN T-cell development, purified or thawed Lin⁻ progenitors were cultured on OP9-DL1 stromal cell monolayers (*Schmitt and Zúñiga-Pflücker, 2002*) plated on tissue-culture treated plates (Corning) using standard culture medium [80% αMEM (Gibco), 20% Fetal Bovine Serum (Sigma-Aldrich), Pen-Strep-Glutamine (Gibco), 50 μM β-mercaptoethanol (Sigma)], grown at 37°C in 5% $CO_2$ conditions, and supplemented with cytokines. All in vitro T-cell generation cultures of Bcl11b-YFP/mCh Lin⁻ BM precursors were supplemented with 5 ng/mL Flt-3L (Peprotech) and 5 ng/mL IL-7 (Peprotech), and were sorted after 6 or seven total days of culture following transduction with a retroviral vector expressing CFP 1 day prior (*Figures 2*, *4F* and *5A*, and *Figure 2—figure supplement 1*, *Figure 4—figure supplements 1–2*). Lin⁻ fetal liver precursors were cultured with 5 ng/mL Flt-3L and 1 ng/mL IL-7 for the indicated number of days before analysis or sorting. For experiments in which *Bcl11b* locus activation was compared in the presence and absence of Notch signals, DN2 progenitors were cultured in parallel with OP9-DL1 stroma and with OP9-Control (without DL1 expression), respectively, as previously described (*Kueh et al., 2016*).

Sorted thymocytes (*Figures 3D* and *6A*), BM-derived DN2 progenitors (*Figure 5A*), and FL-DN progenitors (*Figure 6B*) were seeded manually onto 6000 OP9-DL1 or OP9-Control feeder cells per well in 96-well plates, cultured in standard medium supplemented with 5 ng/mL Flt-3L and either 5 ng/mL IL-7 (BM) or 1 ng/mL IL-7 (Thymocytes and FL), and harvested for analysis after the indicated number of days.

## Flow cytometry and cell sorting

Unless otherwise noted, flow cytometry analysis and fluorescent-activated cell sorting of all *in vitro* and *ex vivo* lymphocytes were prepared using the procedures outlined. Briefly, cultured cells on tissue culture plates and primary cells from lymphoid organs were prepared as single cell suspensions, incubated in 2.4G2 Fc blocking solution, stained with respective surface cell markers as indicated (see *Supplementary file 1*, Key Resources Table), resuspended in HBH, filtered through a 40 µm nylon mesh, and analyzed using a benchtop MacsQuant VYB flow cytometer (Miltenyi Biotec, Auburn, CA) or sorted with Sony Synergy Sorter (Sony Biotechnology, Inc, San Jose, CA). Both instruments contain capabilities to detect mCherry fluorescence by 561 nm laser excitation. All antibodies used in these experiments are standard, commercially available monoclonal reagents widely established to characterize immune cell populations in the mouse; details are given in *Supplementary file 1*. Acquired flow cytometry data were all analyzed with FlowJo software (Tree Star).

## Timelapse imaging

Timelapse imaging of live-cells was used to study *Bcl11b* gene expression dynamics in single cells (*Figures 2* and *4F*, and *Figure 2—figure supplement 1*, *Figure 4—figure supplements 1–2*, and *Video 1*). To prepare for multi-day imaging, PDMS micromesh arrays (250 µm hole diameter, Microsurfaces, AU) containing small microwells that prevent seeded cells from migrating out of a single imaging field of view on 40x objective were adhered to 24-well glass-bottomed plates (Mattek, Ashland, MA). To prevent overcrowding in microwells and enable proper cell tracking, non-GFP expressing OP9-DL1, described in *Kueh et al., 2016*, and sorted CFP+ DN2 progenitors were plated at appropriate densities to achieve ~8 cells/microwell and ~1 cell/microwell, respectively. Cells were cultured in standard medium using Phenol Red-free αMEM (Gibco) and supplemented with 5 ng/mL Flt-3L and 5 ng/mL IL-7.

## Image segmentation and analysis

Cells were segmented using image processing workflow implemented in MATLAB (Mathworks, Natick, MA), as previously described in detail (*Kueh et al., 2013*; *Kueh et al., 2016*). Briefly, this workflow involved: (1) Correction for uneven fluorescence illumination, calculated from a fluorescent slide with uniform intensity, followed by background subtraction; (2) Automated cell segmentation, using an Laplacian filter-based edge detection algorithm, followed by exclusion of non-cell objects by size and shape selection. Cell segmentations were then subject to manual inspection, and segmented objects that did not correspond to cells were then eliminated. For each data set, automated segmentation parameters were chosen such that the fraction of incorrectly identified cells was <1% of the total number of segmented cells. To calculate fluorescence intensities for segmented cells, we first calculated average intensity levels for an annulus surrounding the segmented cell, and subtracted this background value from image intensities in the cell interior. This additional subtraction was performed to remove auto-fluorescence contributions from OP9-DL1 feeder cells to intensity measurements. Fluorescence intensity measurements were either displayed for clonal cell lineages confined within individual microwells (*Figure 2B, C* and *Figure 2—figure supplement 1A*, *Figure 4—figure supplement 2B*), or in a two-dimensional heat map showing the intensity distributions for different indicated time windows for all 218 microwells in a single imaging experiment (*Figures 2B–D* and *4F*, and *Figure 4—figure supplement 1*).

To obtain the time evolution of *Bcl11b* population fractions, we fit the 2D histograms of Bcl11b-YFP and Bcl11b-RFP levels, given by *y* and *r* respectively, to a sum of four 2D Gaussian functions:

$$F(r,y) = \sum_{i=1}^{4} f_i(r,y).$$

Each 2D Gaussian function is given by:

$$f_i(r,y) = \frac{N_i}{2\pi\sigma_{r,i}\sigma_{y,i}\sqrt{1-\rho_i^2}} \cdot \exp\left(-\frac{1}{2(1-\rho_i^2 r^2)}\left[\frac{(r-\mu_{r,i})^2}{\sigma_{r,i}^2} + \frac{(y-\mu_{y,i})^2}{\sigma_{r,i}^2} + \frac{2\rho_i(r-\mu_{r,i})(y-\mu_{r,i})}{\sigma_{r,i}\sigma_{y,i}}\right]\right)$$

Here, we define the four populations $i = 1 \ldots 4$ to correspond to the non-expressing, yellow

mono-allelic, red mono-allelic and bi-allelic populations respectively (*Figure 4—figure supplement 1*). Here, $N_i$ corresponds to the volume under the Gaussian function when integrated over $r$ and $y$. When fitted to experimental 2D histogram, $N_i$ provides an estimate of the number of cells within a given population.

We performed our fitting in two steps: (1) we first obtained the means, standard deviations and correlation coefficients of the *Bcl11b* non-expressing population $(\mu_{r,1}, \ \mu_{y,1}, \sigma_{r,1}, \sigma_{y,1}, \rho_1)$ by fitting the first 2D Gaussian function $f_1(r, y)$ to a 2D histogram of Bcl11b-YFP and Bcl11b-RFP levels of cells at the beginning of the experiment (within the time window $0<t<10$ hr), which were sorted to have both *Bcl11b* alleles inactive. This fit to the initial non-expressing *Bcl11b* population is crucial for accurate determination of different *Bcl11b* allelic populations using the 4-Gaussian fit approach. (2) To obtain a time series of *Bcl11b* allelic population fractions from single-cell data, we then fit the constrained 4-Gaussian model $F(r, y)$ to 2D histograms of Bcl11b-YFP and RFP levels obtained across successive time bins, fixing the parameters for the first Gaussian, and allowing the means, standard deviations and correlation coefficients of the remaining Gaussians $(\mu_{r,j}, \ \mu_{y,j}, \sigma_{r,j}, \sigma_{y,j}, \rho_j)$, j = 2....4 to vary within bounds set by the observed fluorescence distributions of the *Bcl11b* allelic-expressing population. From these fits, we then obtain the observed fraction of cells in the $i$th state in a time window centered on time $t$:

$$f_i^{\mathrm{obs}}(t) = N_i(t) / \sum_{j=1}^{4} N_j(t)$$

To estimate confidence bounds, we also perform error analysis to get the confidence bounds for $f_i(t)$, given by:

$$\delta f_i^{\mathrm{obs}} = f_i^{\mathrm{obs}} \sqrt{\left(\frac{\delta N_i}{N_i}\right)^2 + \frac{\sum_{j=1}^{4} \delta N_j^2}{\left(\sum_{j=1}^{4} N_j\right)^2}}$$

Here, $\delta N_i$ represents the error in the estimation of $N_i$ from least squares fitting.

## Model analysis and fitting

Models for *Bcl11b* activation (*Cis*-only, Sequential *trans-cis*, Parallele *trans-cis*; *Figure 4A*, Appendix) were numerically simulated using an ordinary differential equation solver in MATLAB. The predicted time course from these models were fit to experimental data, using a least-squares procedure with the following free parameters: the *cis*- and *trans*- activation rates ($k_C$ and $k_T$ respectively), and the fraction of cells in each *Bcl11b* non-expressing sub-state, constrained to equal one at t = 0 (sequential and parallel *trans-cis* models only). For both yellow/red mono-allelic and bi-allelic expressing populations fractions, there is a clear lag in their rise kinetics of ~15–20 hr (*Figure 4B*), even though all bi-allelic and mono-allelic expressing cells are all already discernible after the earliest measured time interval (~5 hr, *Figure 4—figure supplement 1*, red arrows). This lag in measured population fraction data occurs, because the earliest Bcl11b expressing cells still have fluorescent reporter levels that are very similar to the non-expressing cell populations, and are therefore not detected by the constrained Gaussian feature described above. Thus, to correct for this lag, we introduced detection time delay into the fitting functions for the three *Bcl11b* –expressing populations, as follows:

$$f_i^{'}(t) = f_i(t - \tau_i)$$

Here, $\tau_i$ denotes the time delay in the detection of the $i$th allelic expression state, taken to be the time at which the detected fraction of cells in the $i$th state increases to a value significantly greater than zero. Note that as RFP and YFP differed in their accumulation rates and detection thresholds, they showed different time delays for detection. We evaluated the goodness of each model fit by calculating its reduced chi-squared value, defined as:

$$\chi^2 = \frac{\left[\sum_{i,j} \frac{f_{i,j} - f_{i,j}^{\mathrm{obs}}}{\delta f_{i,j}^{\mathrm{obs}}}\right]}{\mathrm{d.f.}}$$

where the summation is taken over all allelic expressing states $i$ and all time points $j$, subject to the time lag defined above, and the number of degrees freedom (d.f) is defined as the number of fitted data points, minus the number of fitting parameters. To compare whether the sequential or parallel *trans-cis* models provided a significantly better fit to the data compared to the *cis-* only model, we then took the ratio of reduced chi-squared values for the two compared fits (i.e. their *F* values), and evaluated for statistical difference using the *F*-test. Qualitative predictions for perturbing specific reaction steps (*Figure 5—figure supplement 1*) were obtained by performing a series of simulations with increasing magnitude of perturbation to the same ending time point. In accordance with experimental observations showing some inactivation of the *Bcl11b* locus upon Notch withdrawal (*Figure 5A,B*), perturbations involved both a reduction of the forward rate constant, and an increase in the rate of a reverse reaction, together with a graded attenuation in the perturbation after activation of one *Bcl11b* allele (see *Figure 5—figure supplement 1*, and Appendix for a comprehensive description). Parameters were chosen based on the best fits of the unperturbed time course (*Figure 4B*), although the direction of the predicted shifts in phase space do not depend on the exact parameters being chosen (*Figure 5C,D*, and *Figure 5—figure supplement 1*).

To generate predictions for allelic state distributions from single clones (*Figure 4D,E*), we performed Monte-Carlo simulations of clonal single proliferating progenitor lineages, using Markov transition probabilities determined by best-fit rate constants to sequential or parallel trans-cis models (see Appendix). Here, the cell division time was taken to be 20 hr, corresponding to rates of cell expansion observed in experiments, and measurements of clonal allelic distributions were taken at 100 hr (i.e. after five cell divisions), also matching the time of experimental sampling. Probabilities per cell division for each transition were obtained by converting the continuous-time models to a discrete Markov chain, and these probabilities were taken to be independent between two daughters of the same cell, consistent with the first-order kinetics of these transitions in our models. To test experimental data against each model, we obtained the expected probability of having clones with dual-allelic expression together with mono-allelic expression from two alleles (Y + R + D) or from a single allele (Y + D and R + D) clones, for each model, and compared the observed frequencies from clonal lineage data using a chi-squared test.

## Radiation chimeras

Fetal liver precursor transplanted CD45.1 chimeras were generated to study the long-term T-cell potential of cells without *Bcl11b* enhancer in mice. Individual fetal liver whole organ suspensions were thawed and split for depletion protocols indicated above or stimulated in standard medium supplemented with 50 ng/mL IL-6 (eBioscience), 50 ng/mL SCF (eBioscience), and 20 ng/mL IL-3 (eBioscience) for 2 days to enrich for hematopoietic stem cell (HSC) progenitors. CD45.1 C57BL/6 mice were subjected to sublethal radiation of 1000 rads from a cesium source. Cells were re-suspended in PBS and $10^6$ cells in a volume of 200 μL were injected retro-orbitally into anesthetized, irradiated mice using 31G, 6 mm insulin syringes (BD). Comprehensive splenocyte analysis was performed on 2-month post-transplantation chimeras by sacrificing mice and harvesting spleen and thymus organs following protocols indicated above (*Figure 3—figure supplement 3*).

## Retroviral transduction on Retronectin-DL1-coated plates

Retroviral particles were packaged by transient cotransfection of the Phoenix-Eco packaging cell line with the retroviral construct and the pCL-Eco plasmid (Imgenex) using FuGENE 6 (Promega). Viral supernatants were collected at 2 and 3 days after transfection and immediately frozen at −80°C until use. To infect BM-derived T-cell progenitors, 33 μg/mL retronectin (Clontech) and 2.67 μg/mL of DL1-extracellular domain fused to human IgG1 Fc protein (*Varnum-Finney et al., 2000*) were added in a volume of 500 μL per well in 24-well tissue culture plates (Costar, Corning) and incubated overnight. Viral supernatants were added next day into coated wells and spun down at 2000 rcf for 2 hr at room temperature. BM-derived T-cell progenitors used for viral transduction were cultured for 5 days according to conditions described above, disaggregated, filtered through a 40 μm nylon mesh,

and $10^6$ cells transferred onto each retronectin/DL1-coated virus-bound 24-well supplemented with 5 ng/mL SCF (Peprotech), 5 ng/mL Flt3-L, and 5 ng/mL IL-7.

## Quantification and statistical analysis

The sample size for each experiment, and number of independent experiments are stated in the Figures and Figure Legends. In *Figure 4B*, the best fits of the different models were evaluated by comparing sum-squared errors using the *F*-test (*Figure 4C*), adjusted for different degrees of freedom for each model. A chi-squared test was applied to compare experimental data against model predictions shown in *Figure 4F*. Data that had a calculated $p$-value$<0.05$ was considered statistical significant, and exact P-values are reported in the figure legends. Bar chart data shown (*Figure 6A,B*) represent mean and standard deviation.

## Acknowledgements

We thank M Lerica Gutierrez Quiloan for mouse genotyping and maintenance; N Verduzco and I Soto for animal husbandry; RA Diamond, K Beadle, and D Perez for cell sorting. We also thank members of Kueh, Rothenberg and Elowitz labs for feedback, and T Mitchison for valuable discussions. We also thank Sandy Nandagopal, Pulin Li, Zeba Wunderlich and Nick Pease for comments. This work was funded by an NIH K99/R00 Award (5R00HL119638), a Tietze Foundation Stem Cell Scientist Award, and a CRI/Irvington Postdoctoral Fellowship (to HYK); NIH grants R01AI095943, R01AI083514, and R01HL119102 (to EVR), California Institute for Regenerative Medicine Bridges to Stem Cell Research (to KKHN); and the Louis A Garfinkle Memorial Laboratory Fund, the Al Sherman Foundation, and the Albert Billings Ruddock Professorship (to EVR).

## Additional information

### Funding

| Funder | Grant reference number | Author |
|---|---|---|
| California Institute for Regenerative Medicine | Graduate Student Award | Kenneth KH Ng |
| Howard Hughes Medical Institute | Investigator | Michael B Elowitz |
| National Institutes of Health | R01AI095943 | Ellen V. Rothenberg |
| National Institutes of Health | R01AI083514 | Ellen V. Rothenberg |
| National Institutes of Health | R01HL119102 | Ellen V. Rothenberg |
| National Institutes of Health | R00HL119638 | Hao Yuan Kueh |
| John H. Tietze Foundation Trust | Stem Cell Scientist Award | Hao Yuan Kueh |

The funders had no role in study design, data collection and interpretation, or the decision to submit the work for publication.

### Author contributions

Kenneth KH Ng, Data curation, Formal analysis, Investigation, Writing—original draft, Writing—review and editing; Mary A Yui, Data curation, Formal analysis, Investigation; Arnav Mehta, Satoshi Hirose, Investigation; Sharmayne Siu, Blythe Irwin, Formal analysis, Investigation; Shirley Pease, Resources; Michael B Elowitz, Ellen V Rothenberg, Conceptualization, Supervision, Funding acquisition, Methodology, Writing—original draft, Writing—review and editing; Hao Yuan Kueh, Conceptualization, Data curation, Formal analysis, Supervision, Funding acquisition, Investigation, Methodology, Writing—original draft, Writing—review and editing

Author ORCIDs

Mary A Yui  https://orcid.org/0000-0002-3136-2181
Michael B Elowitz  https://orcid.org/0000-0002-1221-0967
Ellen V Rothenberg  http://orcid.org/0000-0002-3901-347X
Hao Yuan Kueh  http://orcid.org/0000-0001-6272-6673

## Ethics

Animal experimentation: This study was performed in strict accordance with the recommendations in the Guide for the Care and Use of Laboratory Animals of the National Institutes of Health. Animals were bred and maintained in either the Laboratory Animal Facility of the California Institute of Technology, or that of the University of Washington. Animal protocols were reviewed and approved by the Institute Animal Care and Use Committees (IACUC) of the California Institute of Technology (Protocols #1445 and #1409) and the University of Washington Protocol #4397-01.

## Decision letter and Author response

Decision letter https://doi.org/10.7554/eLife.37851.037
Author response https://doi.org/10.7554/eLife.37851.038

## Additional files

### Supplementary files

• Supplementary file 1. List of antibodies used for magnetic bead protocols, flow cytometry analysis, and sorting. Each antibody specifies the cell populations targeted and their corresponding reference figures.
DOI: https://doi.org/10.7554/eLife.37851.025

• Transparent reporting form
DOI: https://doi.org/10.7554/eLife.37851.027

### Data availability

Imaging data, along with MATLAB image processing scripts have been deposited in github: https://github.com/KuehLabUW/ictrack/ (copy archived at https://github.com/elifesciences-publications/ictrack). Source data for Figs. 2,3,4,5, Figure 3-figure supplements 1,2 and 3 have also been included.

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

## Appendix 1

DOI: https://doi.org/10.7554/eLife.37851.028

### Introduction

We describe a series of dynamical models that aim to clarify the interplay between global (*trans*) and locus-specific (*cis*) mechanisms in the control of *Bcl11b* activation and T-lineage commitment. We first use these models to understand the dynamics of normal *Bc11b* activation in an initial population of DN2 progenitors that are inactive for both *Bcl11b* copies (*Figure 4*). Next, to distinguish between these different models, we will make predictions about their behavior on a clonal lineage level (*Figure 4D–E*), and their responses to perturbations of different activation steps (*Figure 5C-D* and *Figure 4—figure supplement 2*, *Figure 5—figure supplement 1*), which we will test experimentally. In these models, we do not explicitly model the ETP to DN2 transition, as our experiments all start with cells that have already turned on CD25; however, as we discuss below, our analysis of the sequential and parallel *trans-cis* models suggest that some of the molecular events we consider could occur prior to the ETP-DN2 transition. We note that these models are simplified representations of more complex underlying systems, and a full understanding of the dynamics of the complete system will involve additional processes not accounted for here. However, we use these minimal models to constrain experimental data, evaluate the plausibility of broad classes of mechanisms, and provide a starting point for further investigation.

### Simple cis-activation model

In this model, activation of *Bcl11b* involves a single, slow first-order step that takes place in cis, that is on the locus of the *Bcl11b* gene itself. This activation step is controlled independently for two copies of *Bcl11b* in a single cell, and with the same rate constant. Under these assumptions, the fraction of non-expressing, mono-allelic and bi-allelic *Bcl11b* expressing cells evolve over time according to the following dynamical equations:

$$\frac{dn_0}{dt} = -2k_C \cdot n_0 \tag{1}$$

$$\frac{dn_y}{dt} = k_C \left( n_0 - n_y \right)$$

$$\frac{dn_r}{dt} = k_C \left( n_0 - n_r \right)$$

$$\frac{dn_{yr}}{dt} = k_C \left( n_r + n_y \right)$$

Here $k_C$ is the first-order rate constant of the slow *cis*-acting step on the *Bcl11b* locus. In our experiments, starting DN2 progenitors were sorted to have no *Bcl11b* expression on either copy. Thus, in our model fitting, we take all starting cells to be in a non-expressing state, following this initial condition:

$$n_0(0) = 1 \tag{2}$$

Accordingly, all other variables are set to zero. Following this initial condition, we performed least-squares fitting, varying $k_c$ to provide the best fit to experimental data (*Figure 4B*). We note that in this and subsequent fits, experimental data were shifted by a fixed time lag, to account for delays in the appearance of fluorescent protein expression.

As seen from best least-squares fit, this model is a poor description of the experimentally observed dynamics of *Bcl11b* activation from DN2 (*Figure 4B–C*): this is because the fraction of bi-allelic expressing cells increases more slowly compared to that of the mono-allelic expressing cells at the earliest time points. To see how this this time lag arises, we can solve for this model analytically, to derive the following solutions:

$$n_0(\tau) = e^{-2\tau} \tag{3}$$
$$n_y(\tau) = e^{-\tau} - e^{-2\tau}$$
$$n_r(\tau) = e^{-\tau} - e^{-2\tau}$$
$$n_{yr}(\tau) = 1 + e^{-2\tau} - 2 \cdot e^{-\tau} \tag{4}$$

Where $\tau = k_C t$ is time in non-dimensional units. At early time points, where $\tau \ll 1$, we can expand these solutions using a power series to obtain:

$$n_y \approx \tau \tag{5}$$
$$n_{yr} \approx \tau^2 \approx n_y^2 \tag{6}$$

At early time points, the fraction of bi-allelic expressing cells is approximately the square of fraction of the mono-allelic expressing cells, and would therefore increase at a slower rate relative to mono-allelic expressing cells.

## Sequential trans-cis activation model

In this model, two rate-limiting steps are required for activation of *Bcl11b*, a *trans*-acting step, which occurs in the nucleus away from the *Bcl11b* locus, and a *cis*-acting step, which occurs independently on each Bcl11b locus. The *trans* step precedes, and is necessary for, the *cis* step. Such a model could describe a reaction scheme, where an initial limiting step, occurring away from the *Bcl11b* locus, activates a regulatory factor that facilitates the *cis*- activation step in a permissive fashion. This regulatory factor could be a chromatin-modifying enzyme, a transcription factor, or any other protein that serves to enable locus remodeling. We note that in this model, it is possible that the *trans*-acting step occurs before the DN2 transition (*Figure 4A*, gray arrows).

There are five states, a *trans*-inactive state $M_0$, where this *trans* factor is absent, and four states, $N_0$, $N_y$, $N_r$, and $N_{ry}$, where the *trans* factor is present, and two copies of *Bcl11b* exist in either active or inactive states. The time evolution of the fraction of DN2 progenitors in these different states are given by:

$$\frac{dm_0}{dt} = -k_T m_0 \tag{7}$$
$$\frac{dn_0}{dt} = k_T m_0 - 2k_C n_0$$
$$\frac{dn_y}{dt} = k_C (n_0 - n_y)$$
$$\frac{dn_r}{dt} = k_C (n_0 - n_r)$$
$$\frac{dn_{yr}}{dt} = k_C (n_r + n_y)$$

Here, $k_T$ and $k_C$ correspond to the first-order rate constants for the *trans* and *cis*-acting steps respectively. The value of these rate constants were determined using least-squares fitting to experimental data, subject to the constraint that the initial DN2 progenitors that we sorted are all inactive for both copies of *Bcl11b*, and must exist in either *trans*-active or *trans*-inactive states:

$$m_0(0) + n_0(0) = 1. \tag{8}$$

This constraint results in one additional fitting parameter to the model. The best fit trajectory is shown in the main text (*Figure 4B*), and best-fit parameters are shown in *Appendix 1—table 1*. Unlike the simple *cis*-activation model, this model can give rise to a rise in the fraction of bi-allelic expressing fraction concurrent with the mono-allelic expressing fraction (*Figure 4B–C*); thus, from least-squares fitting of experimental data alone, the sequential activation model can plausibly explain experimentally observed population dynamics.

**Appendix 1—table 1.** Best fit parameters of the sequential *trans-cis* activation model to data, with 95% confidence intervals.

| Parameter | Units | Best-fit | Lower bound | Upper bound |
|---|---|---|---|---|
| $k_c$ | 1/hr | $3.5 \times 10^{-2}$ | $3.3 \times 10^{-2}$ | $3.6 \times 10^{-2}$ |
| $k_t$ | 1/hr | $3.3 \times 10^{-2}$ | $3.1 \times 10^{-2}$ | $3.6 \times 10^{-2}$ |
| $m_0$ | l | 0.81 | 0.78 | 0.83 |

DOI: https://doi.org/10.7554/eLife.37851.029

## Parallel trans-cis-activation model

In this model, *cis*-acting and *trans*-acting steps are also required for activation of *Bcl11b*, similar to the sequential activation model. However, in contrast to the sequential model, *cis* and *trans* steps occur in parallel with each other, such that they occur in either order. In this model, the *trans* step could represent activation of a *trans* factor necessary for transcription of a *cis*-activated locus. For instance, the *trans*-acting step could correspond to the activation of a factor that promotes the polymerase recruitment.

In this model, there are four *trans*-inactive states $M$ and four *trans*-active states $N$, each corresponding to different states of locus activation. The time evolution of the fraction of cells in these states are given by:

$$\frac{dm_0}{dt} = -2k_C m_0 \tag{9}$$

$$\frac{dm_y}{dt} = k_C m_0 - (k_T + k_C)m_y$$

$$\frac{dm_r}{dt} = k_C m_0 - (k_T + k_C)m_r$$

$$\frac{dm_{yr}}{dt} = k_C(m_r + m_y) - k_T m_{ry}$$

$$\frac{dn_0}{dt} = k_T m_0 - 2k_C n_0$$

$$\frac{dn_y}{dt} = k_C n_0 + k_T m_y - k_C n_y$$

$$\frac{dn_r}{dt} = k_C n_0 + k_T m_r - k_C n_r$$

$$\frac{dn_{yr}}{dt} = k_C(n_r + n_y) + k_T m_{ry}$$

Here, $k_T$ and $k_C$ correspond to the first-order rate constants for the *trans-* and *cis-* acting steps. As experiments start with cells that do not express *Bcl11b*, the following constraint describes the fitting of our models:

$$m_0(0) + m_y(0) + m_r(0) + m_{ry}(0) + n_0(0) = 1 \tag{10}$$

This constraint results in four additional free parameters to the least-squares fit. Upon performing a least-square fit to experimental data, we find that this model also recapitulates the early rise in the fraction of bi-allelic expressing cells, as observed in the data (***Figure 4B–C***; see ***Appendix 1—table 2*** for best-fit parameter values). Of note, our model fit suggests that a significant fraction of DN2 progenitors may already exist in a state where one or both *Bcl11b* alleles are already activated in *cis* (***Appendix 1—table 2***). This feature of our fit will enable us to distinguish between the sequential and parallel activation models using clonal lineage data, as we discuss further below.

**Appendix 1—table 2** Best fit parameters of the parallel *trans-cis* activation model to data, with 95% confidence intervals.

| Parameter | Units | Best-fit | Lower bound | Upper bound |
|-----------|-------|----------|-------------|-------------|
| $k_c$ | 1/hr | $5.2 \times 10^{-3}$ | $4.3 \times 10^{-3}$ | $6.2 \times 10^{-3}$ |
| $k_t$ | 1/hr | $2.7 \times 10^{-2}$ | $2.6 \times 10^{-2}$ | $2.8 \times 10^{-2}$ |
| $m_0$ | (fraction) | 0.21 | 0.14 | 0.33 |
| $m_r$, $m_y$ | (fraction) | 0.29 | 0.27 | 0.31 |
| $m_{ry}$ | (fraction) | 0.21 | 0.20 | 0.23 |

DOI: https://doi.org/10.7554/eLife.37851.030

## Comparative analysis of sequential and parallel trans-cis activation models

### Clonal heterogeneity analysis

So far, both sequential and parallel *trans-cis* activation models provide a reasonable fit to the population dynamics of mono-allelic and bi-allelic cell fractions starting from non-expressing progenitors (*Figure 4B–C*). How can we further distinguish between these two models? So far, we have only considered predictions based on the behavior of whole cell populations; however, analysis of correlations within individual lineage trees can allow discrimination of distinct dynamic mechanisms, as was demonstrated in recent work and in classic experiments (*Luria and Delbrück, 1943*; *Hormoz et al., 2016*; *Blanpain and Simons, 2013*). As these two models differ in activation state trajectories taken during *Bcl11b* activation, they would be expected to generate distinct distributions of allelic activation states in single clonal lineages.

To derive *Bcl11b* activation state distributions expected from either sequential or parallel activation models, we first reformulate these models (*Equations 7 and 9*) as discrete time Markov Chains (*Gardiner, 2009*), where each time step represents a single cell cycle. First, let $N$ be the total number of states. Next, define a random state variable $S_t$, corresponding to the state of the cell at the number $t$. For the sequential model (*Equation 7*), the list of states is $\{m_0, n_0, n_y, n_r, n_{yr}\}$, with $N = 5$; for the parallel model (*Equation 9*), the list of states is $\{m_0, m_y, m_r, m_{yr}, n_0, n_y, n_r, n_{yr}\}$, with $N = 8$. In our descriptions below, we will enumerate all the states as $i = 1...N$ in such a specified order.

Next, we define $T$, a transition matrix with $N \times N$ elements, where $T_{ij}$ represents the probability of a cell transitioning from state $j$ to state $i$ in a single cell cycle. For a given cell cycle time $t_c$, we can solve the differential equations in *Equations 7 and 9* to obtain corresponding transition probabilities, that is $T_{ij}(t_c)$. In our simulations, we first solve for these transition probabilities, using the best-fit rate constants in *Appendix 1—table 1* and *Appendix 1—table 2*. These experiments also used a cell cycle time of $t_c = 20$ hr. This was chosen in accordance with the amount of cell expansion observed in imaging experiments, though our conclusions are not expected to depend on the exact value of the cell division time. With this transition probability matrix, we can then simulate state transitions across a lineage of dividing cells, according to the following formula:

$$\Pr(S_{t+1} = i | S_t = j) = T_{ij} \tag{11}$$

Here $S_t$ represents the state of the cell at the ($t$)th cell cycle. In the Monte-Carlo simulations, each cell gives rise to two cells at each cell division, and each daughter cell chooses its fate randomly and independently from its sibling, based on this formula. This process is repeated iteratively for every descendant from a single ancestor until a designated stopping time (5 cell cycles, or 100 hr, corresponding to the end of the imaging experiment), whereby a complete lineage tree is generated.

From these clonal lineage simulations, we find the sequential and parallel *trans-cis* activation models yield divergent predictions of heterogeneity in *Bcl11b* allelic activation at the level of single clones. For the sequential activation model, non-expressing ancestors predominantly generate a mixture of progeny with mono-allelic expression from both *Bcl11b* copies prior to bi-allelic *Bcl11b* activation (*Figure 4D–E*). While some clones only express a

single specific *Bcl11b* copy prior to bi-allelic activation, these clones were rare relative to those with mono-allelic expression from each of the two alleles (*Figure 4D*). This is because all non-expressing progenitors still have both *Bcl11b* copies in a *cis*-inactive state; thus, upon cell division, all daughters of a non-expressing parent retain the same probability of activating either allele.

By contrast, the parallel *trans-cis* activation model gave rise to a large frequency of clones with mono-allelic expression from only one specific allele (*Figure 4D–E*), either red or yellow, varying between, but not within, different clones. This reflects the accumulation of non-expressing progenitors that have a single *Bcl11b* copy present in an open state, but lack the *trans*-acting factors necessary to induce expression from this opened locus (58% total, *Appendix 1—table 2*). These clones pass through a single specific mono-allelic activation intermediate prior to bi-allelic activation. Additionally, in the parallel activation model, a small percentage of clones transition directly to a bi-allelic expressing state without first passing through a mono-allelic state (*Figure 4D–E*), a behavior that does not occur for the sequential activation model. This 'tunneling' of non-expressing cells to a bi-allelic expression state reflects the existence of non-expressing cells with both alleles open that still lack the critical *trans*-acting step to enable their expression. For these cells, activation of the *trans*-acting step causes both alleles to turn on simultaneously.

In our experimentally observed distributions of allelic activation states, we found that individual clones predominantly showed mono-allelic expression from only one allele (*Figure 4F*, 7/9 clones observed), but only rarely showed mono-allelic expression of both alleles (*Figure 4F*, 1/9 clones observed). This distribution of single-specific mono-allelic clones was significantly different from the fractions predicted for the sequential activation model ($p<0.01$, $\chi^2 = 6.8$, d.f. = 1), but not significantly different from predictions for the parallel activation model.

Furthermore, the experimentally observed distributions also showed evidence for simultaneous activation of both alleles from a non-expressing state (*Figure 4E*, 1/9 clones), consistent with the occurrence of a parallel *trans*-activation event in a DN2 progenitor with both *Bcl11b* alleles pre-activated in *cis*, which is only allowed in the parallel activation model. Taken together, the experimental clonal lineage data favor the parallel activation model as an explanation for the underlying kinetic processes controlling *Bcl11b* activation, suggesting that the *trans*- acting step necessary for *Bcl11b* activation occurs in parallel with the *cis*-level step on the *Bcl11b* locus.

## Effects of perturbation of *cis* and *trans* activation steps

To further discriminate between sequential and parallel *trans-cis* activation events, and to gain insights into the molecular mechanisms underlying control of the *trans*-acting step, we analyzed the predicted effects of perturbing different reaction steps for each model. We then tested these predictions by removing the Notch signaling ligand DL1, an essential T-cell developmental signal that controls *Bcl11b* activation probabilities. Here, we show that perturbations of the reaction steps in different models generate distinct shifts in the distribution of mono-allelic and bi-allelic expressing cells, which can be compared to experiments for model discrimination. In this simulation analysis (*Figures 5C-D* and *Figure 5—figure supplement 1*), we perturbed both *cis*- and *trans*-acting steps in the two models in the same way, by reducing its forward rate while introducing a non-zero backward rate. This assumption of reversibility reflects our previous observations that *Bcl11b* can turn back off in a small fraction of cells. We previously noted that although Bcl11b expression maintenance rapidly becomes Notch-independent, there is a small percentage of cells that can lose Bcl11b expression again shortly after activation, if Notch signaling is removed (*Kueh et al., 2016*). Thus, building on this observation, we reduced the forward rate constant by a fraction $d$, while concomitantly increasing the back rate constant by the same amount $d$. Also, in accordance with experimental observations (We attenuated Notch-dependency as described because experiments showed that cells with both *Bcl11b* alleles active show a reduced rate of reversion to an inactive state upon Notch signaling withdrawal. The molecular basis for this attenuation in Notch dependency is currently unclear, but likely

involves involve a parallel process occurring in the nuclei of progenitors to stabilize a Notch-driven T-lineage program over time.), the effect of each perturbation on the change in rate constants was further reduced a multiplicative factor $f(<1)$ for transitions to and from a dual-allele expressing state. The perturbed rate constants are labeled in the state transition diagrams in *Figure 5—figure supplement 1*, and their definitions, as described here, as listed in *Appendix 1—tables 3–6*. We note that, while the magnitudes of the experimentally observed shifts depend on these chosen values, the *directions* of these shifts in phase space upon perturbation - corresponding to the increases or decreases in ratio of bi-allelic to mono-allelic expressing cells - are not dependent on the specific values of chosen constants, and thus represents a robust qualitative prediction of the modeling.

By numerically simulating these models, we found that different perturbations generated distinct shifts in *Bcl11b* mono-allelic to bi-allelic ratios that could then be used to distinguish between effects of Notch on the *cis* versus *trans* steps. Specifically:

- When *cis*-acting steps are perturbed in both the sequential and parallel activation models, non-expressing or mono-allelic expressing starting progenitors reach a final state with reduced bi-allelic expression, and either reduced (sequential activation model) or increased (parallel activation model) mono-allelic expression (*Figures 5C* and *Figure 5—figure supplement 1*, blue arrows). However, the ratio of bi-allelic to mono-allelic expressing cells $(F_b/F_m)$ invariably decreases, such that the line connecting initial to final states in phase space makes a smaller angle with the *x*-axis when perturbation is applied. This result does not depend on the exact perturbation strengths specified by *d*, and also does not depend on whether perturbations are reduced for transitions involving the bi-allelic state ($f<1$), or whether they remain the same ($f = 1$) (See *Figure 5—figure supplement 1*).
- When the *trans*-acting step in the sequential activation model is perturbed, progenitors starting without *Bcl11b* expression (*Figure 5—figure supplement 1*, sequential model) reach a final state with a reduced fraction of mono-allelic and bi-allelic expressing cells. Progenitors starting with mono or bi-allelic expression are not affected (*Figure 5—figure supplement 1*, sequential model).
- When the *trans*-acting step in the parallel activation model is perturbed, non-expressing and mono-allelic progenitors reach a final state with reduced mono-allelic and bi-allelic expression, but also show a decrease in the *ratio* of mono-allelic to bi-allelic expressing cells (*Figure 5D*). As explained above, this increase in $F_b/F_m$ cannot occur with inhibition of the *cis*-acting step in either the sequential or parallel model, and cannot occur when starting with mono-allelic expressing cells in the sequential activation model. Hence, this shift distinguishes the parallel from the sequential activation model. Here, we note that bi-allelic cells show a proportionately smaller decrease relative to mono-allelic cells, because transitions involving this state are impacted less by the perturbation ($f<1$). If all transition rates were affected uniformly ($f = 1$), both mono-allelic and bi-allelic cells would be affected similarly, such that the ratio of the two populations would remain the same (*Figure 5—figure supplement 1*).
- Furthermore, when starting from a bi-allelic expressing state, perturbation of the *trans* step causes direct transition to a non-expressing state, without passage through a mono-allelic expressing intermediate (*Figure 5—figure supplement 1*, parallel model). This bi-allelic inactivation represents the reversion of the progenitor to a state where cells still maintain two *cis*-active *Bcl11b* alleles *cis*, but have now inactivated a parallel *trans*-acting step necessary for expression from a *cis*-opened locus. As this bi-allelic shutoff would not be predicted to happen upon perturbation of any other step in either model, it provides an additional signature of the parallel activation model.

Taken together, this analysis suggests that the sequential and parallel activation models could potentially be distinguished by analyzing changes in the fractions of non-expressing, mono-allelic, and bi-allelic cells in response to perturbation, if this perturbation involved a disruption of the *trans*-acting step for *Bcl11b* expression.

To test these predictions, we sorted DN2 progenitors with different numbers of active *Bcl11b* alleles, cultured them *in vitro* in either the presence (unperturbed condition) or

absence (perturbed condition) of Notch signaling, and then analyzed allelic activation states after four days using flow cytometry.

Consistent with parallel activation model, Notch withdrawal reduced the proportion of mono-allelic to bi-allelic expression in cells that started with zero or one active copies of *Bcl11b* (**Figure 5A–B**). It also caused the direct transition of bi-allelic expressing cells to a non-expressing state, without passing through a mono-allelic intermediate state, as can be seen in the flow cytometry analysis of the effects of Notch removal on progenitors expressing both copies of *Bcl11b* (**Figure 5A**, green arrow). These experiments reveal a strong correlation at the single-cell level between expression levels of Bcl11b-YFP and Bcl11b-mCherry alleles, in cells that are shutting off their expression, suggesting that the inactivation of *Bcl11b* upon Notch withdrawal occurred in a highly synchronous manner for two alleles.

Taken together, our results support the parallel activation model over sequential activation model, and indicate that Notch signaling effectively represents one parallel *trans*-acting step necessary for *Bcl11b* expression.

**Appendix 1—table 3.** Perturbing the *cis*-acting step in the sequential activation model.

| Parameter | Description |
|---|---|
| $k_f^0 = k_C(1-d)$ | *cis*-activation rate, from non-expressing to mono-allelic state |
| $k_f^1 = k_C(1 - f \cdot d)$ | *cis*-activation rate, from mono-allelic to bi-allelic state |
| $k_r^0 = k_C \cdot d$ | back *cis*-activation rate, from mono-allelic to non-expressing state |
| $k_r^1 = k_C \cdot f \cdot d$ | back *cis*-activation rate, from bi-allelic to mono-allelic state |
| $k_T$ | *trans*-activation rate |
| $d$ | 0 to 0.35 |
| $f$ | 0.4 |

DOI: https://doi.org/10.7554/eLife.37851.031

**Appendix 1—table 4.** Perturbing the *trans*-acting step in the sequential activation model.

| Parameter | Description |
|---|---|
| $k_C$ | *cis*-activation rate |
| $k_f^0 = k_T(1-d)$ | *trans*-activation rate |
| $k_r^0 = k_T \cdot d$ | back *trans*-activation rate |
| $d$ | 0 to 0.35 |

DOI: https://doi.org/10.7554/eLife.37851.032

**Appendix 1—table 5.** Perturbing the *cis*-acting step in the parallel activation model.

| Parameter | Description |
|---|---|
| $k_f^0 = k_C(1-d)$ | *cis*-activation rate, from non-expressing to mono-allelic state |
| $k_f^1 = k_C(1 - f \cdot d)$ | *cis*-activation rate, from mono-allelic to bi-allelic state |
| $k_r^0 = k_C \cdot d$ | back *cis*-activation rate, from mono-allelic to non-expressing state |
| $k_r^1 = k_C \cdot f \cdot d$ | back *cis*-activation rate, from bi-allelic to mono-allelic state |
| $k_T$ | *trans*-activation rate |
| $d$ | 0 to 0.35 |
| $f$ | 0.4 |

DOI: https://doi.org/10.7554/eLife.37851.033

**Appendix 1—table 6.** Perturbing the trans-acting step in the parallel activation model.

| Parameter | Description |
|---|---|

*Appendix 1—table 6 continued on next page*

*Appendix 1—table 6 continued*

| Parameter | Description |
| --- | --- |
| $k_c$ | *cis*-activation rate |
| $k_f^0 = k_T(1-d)$ | *trans*-activation rate, from non-expressing/mono-allelic state |
| $k_f^1 = k_T(1 - f \cdot d)$ | *trans*-activation rate, from bi-allelic state |
| $k_r^0 = k_T \cdot d$ | back *trans*-activation rate, from non-expressing/mono-allelic state |
| $k_r^1 = k_T \cdot f \cdot d$ | back *trans*-activation rate, from bi-allelic state |
| $d$ | 0 to 0.65 |
| $f$ | 0.2 |

DOI: https://doi.org/10.7554/eLife.37851.034

