## [Decision Letter]

Thank you for submitting your article “A stochastic epigenetic switch controls the dynamics of T-cell lineage commitment”. Your article has been reviewed by three peer reviewers, and the evaluation has been overseen by a Reviewing Editor and a Senior Editor. We find your work to be of strong interest for publication in *eLife* after suitable revision. Key issues that should be addressed in your revised text are found in the critiques below. In particular, the mathematical modeling and your interpretation of the in vivo and in vitro results (see comments below) require a more thorough as well as nuanced elaboration with suitable discussion of the limitations. Importantly, no additional experimental work is being requested by us.

*Reviewer 1:*

This manuscript aims at addressing the interplay between genetic and epigenetic regulation in thymocyte development. More specifically, the dynamics of expression of *Bcl11b* (a key transcription factor for T cell differentiation). The study relies on a two-color scheme whereby two BAC are engineered to express *Bcl11b* with distinguishable fluorescent reporter (with or without editing of enhancer regions). These constructs enable the authors to readily separate *trans*-effect (e.g. driven by the expression of transcription factors) from *cis*-effect (e.g. driven by chromatin changes and epigenetic regulation at the level of each individual chromosome). This experimental tour-de-force is novel and provides very striking quantitative results about the dynamics and heterogeneity of gene transcription in primary cells.

However, there are three experimental features that the authors fail to account for in their modeling effort. First, the time dynamics of the bi-allelic expression of *Bcl11b* seems to plateau around 20hr, while the mono-allelic graphs are still rising (Figure 4A). Second, the heterogeneity in bi-allelic gene expression appears to differ between stage DN2B and stage DN3 (going from very heterogeneous to very correlated). Third, there is a long time delay in bi-allelic expression, with a sharp increase around 20hr. All observations do not jive well with the "simple / stationary" biochemical models proposed in this study.

Overall, the model fitting in the paper is a bit underwhelming: none of the proposed models clearly recapitulates the qualitative dynamics of bi-allelic expression in Figure 4A. The authors do convincingly rule out a *cis*-only model (the independence of each allele would not allow for high frequency of bi-allelic expression); yet, both sequential *trans-cis* and parallel *trans-cis* do not capture the plateau in bi-allelic expression (a fit of the second moment is not attempted: it could have emphasize the discrepancy between model and experimental data).

There exist better statistical tests that the authors should apply to improve on model assessment. The authors do not describe sufficiently how they compute the p significance of their model.

A proper handling of the Chi-square test would require the accounting of experimental errors in order to assess whether the fitted models is within statistical bounds. Moreover, a Box-Jenkins test would demonstrate that the stationary *trans-cis* models fail to capture the richer aspect of the experimental measurements (current models yield strongly time-correlated residuals). Both statistical tests would most likely reject the present model and give license to the authors to explore more complex models of gene regulation of these *Bcl11b* construct.

One overlooked aspect and complication for the system under study is that the expression of *Bcl11b* over long timescales (>10 hr) must drive the thymocytes to differentiate and to alter their genetic/epigenetic regulation. In the context of explaining the transition towards bi-allelic expression, this differentiation may imply a slowing down of the second step. In other words, one could have one rate *k_C_* to transition to mono-allelic expression, and a smaller rate *k'_C_* to transition from mono- to bi-allelic expression. This might tackle the plateau but still would not explain the time delay for bi-allelic expression. Again, I would encourage the authors to explore more complex models using better statistical tests: this would enable them to, at least, propose better accounting of their experimental data. Depending on what model improves on data fitting, further experimental validation may or may not be warranted (it might be beyond the scope of the present work).

Overall, this is a very interesting study presenting new quantitative observations about the dynamics of *Bcl11b* expression in developing thymocytes. I would encourage the authors to improve their modeling to deliver a better understanding of the competition between *cis*- and *trans*-effect for gene regulation at the chromosome level.

*Reviewer 2:*

In this study the authors engineered a dual-color reporter mouse, where the two *Bcl11b* copies were tagged with distinct fluorescent proteins followed by live cell imaging in order to examine *Bcl11b* dynamics in T cell progenitor cells. The authors use multiple approaches, including perturbation experiments to identify the contributions of *cis*- and *trans*-acting inputs that modulate *Bcl11b* expression. The analyses show that intrinsically stochastic events that occur at single *Bcl11b* alleles dictate the timing and outcome of T lineage cell fate decisions.

This is an elegant study revealing how a distal regulatory genomic region controls the rate of epigenetic regulation, plausibly involving E-proteins and non-coding transcription, whereas in a separate mode of regulation through a distinct pathway (Notch signaling), *Bcl11b* enhancer activity is induced.

The findings are very interesting. The data indicate a very long activation time constant associated with all-or-none irreversible activities. I think this reflects the repositioning of the *Bcl11b* locus from the lamina to the nuclear interior. The authors kind of discuss this possibility but it could be described more precisely. Along the same line the authors note "As *Bcl11b* turn on, its promoter establishes new contacts with the distal enhancer, resulting in de novo formation of an altered topological domain…". I think it is the other way around.

First there is a change in conformation soon followed by activation of *Bcl11b* expression once the locus has repositioned. Of course I might be mistaken about this but the authors may want to discuss these possibilities in greater detail.

*Reviewer 3:*

In this manuscript, the authors investigate the epigenetic switch controlling activation of *Bcl11b*, a key gene in T-cell fate commitment. In particular, they develop a two-colour labelling approach which allows them to distinguish between *cis* and *trans* regulation. In combination with mathematical modelling, they conclude that *Bcl11b* activation dynamics contains parallel *cis* and *trans* activating steps, with the *cis* steps controlled by a distal enhancer, while the *trans* steps are Notch-dependent. Overall, I found this to be an elegant and very insightful manuscript that makes substantial progress on the central, but little studied, question of *cis* versus *trans* regulatory control in cell fate decision making. The mathematical modelling is also very effectively integrated into the story to allow decisive conclusions to be made on the arrangement of the *cis* and *trans* regulatory logic. I am therefore very much in favour of publication in *eLife* subject to some amendments.

• To my knowledge, the only other system to have been sufficiently studied to say anything in detail about the *cis/trans* logic is the FLC silencing system in Arabidopsis. I think it would be insightful to compare and contrast the two systems in more detail. FLC incorporates a cold-induced upstream *trans* upregulation, followed by two sequential *cis* steps (nucleation and spreading), where again the first *cis* step is very slow and stochastic. This Polycomb silencing system is in contrast to the parallel *cis/trans* activating switch at *Bcl11b*. Clearly there will be variety in the choice of *cis/trans* control and probably these two arrangements are just scratching the surface of the possible switching set-ups. But can the authors nevertheless comment more on the advantages of the particular *trans/cis* circuitry that appears to be implemented at *Bcl11b*? With regard to FLC, I would also cite Yang et al., 2017, as that is the only other paper to my knowledge to implement two colour labelling with the goal of distinguishing *cis/trans* regulation.

• In the section where modelling is used to probe the function of Notch, I found the intuition for why perturbing the *cis/trans* steps changed the ratio of mono- versus bi-allelic expressing cells to be unclear. Could this be made clearer?

*Reviewing Editor:*

How well does the in vitro culture system that images individual DN2 cells as they divide and differentiate in the OP-9 system capture the developmental and gene dynamics of these cells in their thymic niche?

In Figure 1 using their dual *Bcl11b* reporter system the authors show that DN2A progenitors in the mouse thymus are heterogeneous, approx. 30% manifest mono-allelic whereas approx. 40% evidence bi-allelic expression. DN2B cells, in contrast manifest predominantly bi-allelic expression (98%). There are two distinct interpretations of these initial observations. (i) DN2A cells undergo stochastic mono-allelic activation of *Bcl11b* and then go on to activate the second allele. (ii) the DN2A compartment is heterogeneous in its induction of *Bcl11b* with 40% of the cells simultaneously inducing both alleles and 30% of the cells exhibiting mono-allelic expression. The former cells are at a developmental advantage and outcompete the others in the DN2b compartment. If the latter interpretation is correct then it would suggest that in vivo the simultaneous bi-allelic activation path is favored. Furthermore the in vitro system being used to analyze the developmental activation of *Bcl11b* alleles favors the less efficient developmental path that DN2a cells can undertake. Another observation that seems to support this interpretation is that in vivo DN2a cells are suggested to convert into DN2b in three days and this is associated with consolidation of bi-allelic expression of *Bcl11b*. Figure 2D shows that even after 95 hr in culture approx. 60% of the DN2a cells have either failed to induce *Bcl11b* or manifest mono-allelic expression. Thus, it is possible that the in vitro system is biased toward the less efficient or minor developmental trajectory that the cells undertake in the thymus.

---

## [Author Response]

Reviewer 1:[…] However, there are three experimental features that the authors fail to account for in their modeling effort. First, the time dynamics of the bi-allelic expression of Bcl11b seems to plateau around 20hr, while the mono-allelic graphs are still rising (Figure 4A). Second, the heterogeneity in bi-allelic gene expression appears to differ between stage DN2B and stage DN3 (going from very heterogeneous to very correlated). Third, there is a long time delay in bi-allelic expression, with a sharp increase around 20hr. All observations do not jive well with the "simple / stationary" biochemical models proposed in this study.

We thank reviewer 1 for pointing out these features of the population fraction time-course dynamics that were not clearly explained by the model in the original manuscript.

In particular, the initial time lag in appearance of bi-allelic cells, followed by the apparent sharp increase and plateau, were not accounted for in the model. Motivated by reviewer #1’s comments, we re-examined our data and model fitting procedures to determine whether these features reflect actual underlying population dynamics or technical artifacts.

In the original manuscript, to estimate fractions of different *Bcl11b*-expressing cell populations in an unbiased manner (without manually setting thresholds), we fit the two-dimensional histograms of single-cell *Bcl11b*‐YFP and *Bcl11b*‐RFP levels to a combination of four 2D Gaussians, each corresponding to a distinct *Bcl11b* allelic population (see Materials and methods). We then used the best-fit volume under each Gaussian as an estimate of population size. While this fitting procedure provides good estimates of population fractions at later times, it exhibited wide confidence intervals at earliest timepoints (t = 0 to 20 hrs), because the different *Bcl11b*-expressing populations are not yet clearly defined. At these time points, cells have begun to turn on one or both *Bcl11b* alleles (see Figure 4—figure supplement 1, compare 2.5 versus 7.5 hrs, and red arrows), yet they have not yet formed a defined population distinct from the non-expressing population. The resulting overlap among the Gaussians led to poor parameter estimates.

In the new version, we improved the fitting procedure in two ways: First, we constrained the ranges of means and standard deviations for the four Gaussian components to prevent them from overlapping significantly with each other. Second, we employed an iterative fitting procedure, where we first obtained parameters for the *Bcl11b* non-expressing population by fitting data at the initial time point (t = 2.5 hr), and then fixed these parameters when fitting the other three Gaussians in the second stage of multiple component Gaussian fitting. We have described this procedure in detail in the Materials and methods section.

This procedure provided better–constrained estimates of the population fraction dynamics for the red and yellow mono-allelic expressing populations and the bi-allelic populations (Figure 4B), with substantially narrower confidence intervals compared to our previous calculations.

As a result of the new fitting procedure, there is now a lag prior to the rise of both mono-allelic and bi-allelic population fractions, due to exclusion of the earliest mono-allelic and bi-allelic expressing cells (see Figure 4—figure supplement 1, 7.5 hr) by the more stringent fitting constraints, as described above. To account for this time lag in the appearance of these populations, we have incorporated a time delay τ in model fits, which depends on the specific population under investigation but is fixed across different models, and accounts for the delay between the first emergence of Bcl11b expressing cells in the data (Figure 4—figure supplement 1), and its detection through our 2D Gaussian fitting procedure. The lags are now indicated by gray shading in Figure 4B. All other aspects of model fitting remain the same.

This improved fitting procedure leads us to the same conclusion – the sequential or parallel *trans-cis* models fit the data better than the *cis*‐only model. As before, the *cis*‐only model yields a delay in the increase of bi-allelic expressing cells relative to mono-allelic expressing cells that is not seen in the experimental data (Figure 4B, gray line). In contrast, both sequential or parallel model fits can explain the roughly concurrent increase in mono-allelic and bi-allelic expressing cells seen in the data. In the new fits, there is less systematic deviation in the fit residuals, especially for bi-allelic expressing cells at later time points. More complex models incorporating time-dependent changes in switching rates failed to significantly improve upon the goodness of these fits (see point below).

Together, these changes improve the fitting significantly and strengthen the conclusions about the relative goodness of fit of the models considered here, and continue to support our conclusion that the parallel *trans-cis* model remains the most plausible among those considered here.

Overall, the model fitting in the paper is a bit underwhelming: none of the proposed models clearly recapitulates the qualitative dynamics of bi-allelic expression in Figure 4A. The authors do convincingly rule out a cis-only model (the independence of each allele would not allow for high frequency of bi-allelic expression); yet, both sequential trans-cis and parallel trans-cis do not capture the plateau in bi-allelic expression (a fit of the second moment is not attempted: it could have emphasize the discrepancy between model and experimental data).

We appreciate this point, which together with the previous one, motivated the improved analysis described in the response above. This re-analysis shows that the apparent plateau was a result of the poorly constrained *Bcl11b* mono-allelic populations that effectively ‘absorbed’ some of the bi-allelic cells during fitting. With the new procedure, the bi-allelic cell fractions no longer appear to plateau at 20 hours, but instead continue to increase in a more linear manner.

There exist better statistical tests that the authors should apply to improve on model assessment. The authors do not describe sufficiently how they compute the p significance of their model.A proper handling of the Chi-square test would require the accounting of experimental errors in order to assess whether the fitted models is within statistical bounds.

We agree with reviewer #1 and made several changes in response to this comment. First, in re-doing the least squares fitting to obtain population fraction data (see above), we calculated 95% confidence intervals for the estimates of allelic population sizes from fitting, and used error propagation to determine resultant confidence intervals for the population fraction data (see Materials and methods). Second, we used these confidence intervals to re-calculate reduced chi-squared values (shown in Figure 4C), which were then used to evaluate whether the sequential or parallel *trans-cis* models provided a significantly better to the data compared to the *cis*‐only model, by calculating the ratio of reduced chi‐squared values for the two compared fits (i.e. their F values), and evaluating for statistical difference. These details are now described in detail in the Materials and methods section. This analysis further strengthens our main conclusion that both sequential and parallel *trans-cis* models better explain the data compared to the simple *cis*‐only model.

Moreover, a Box-Jenkins test would demonstrate that the stationary trans-cis models fail to capture the richer aspect of the experimental measurements (current models yield strongly time-correlated residuals). Both statistical tests would most likely reject the present model and give license to the authors to explore more complex models of gene regulation of these Bcl11b construct.

After the improved analysis, we no longer observe systematic temporal correlations in the residuals in the bi-allelic time course. We do see some systematic under-fitting (over-fitting) of the yellow (red) mono-allelic expressing population at later time points (t > 35-45 hrs). However, as both mono-allelic populations are expected to rise with the same dynamics, we believe that these discrepancies reflect technical differences in the detection of the two fluorescent proteins, which show different brightness and fold increases over background. Due to these technical limitations, we expect it will be difficult for us to extract further biological conclusions from the finer features of time course data with the present data set. In the future, experimental improvements, such as the use of more sensitive reporter systems, should enable more detailed time-series analysis, but would be beyond the scope of the present paper.

One overlooked aspect and complication for the system under study is that the expression of Bcl11b over long timescales (>10 hr) must drive the thymocytes to differentiate and to alter their genetic/epigenetic regulation. In the context of explaining the transition towards bi-allelic expression, this differentiation may imply a slowing down of the second step. In other words, one could have one rate k_C_ to transition to mono-allelic expression, and a smaller rate k'_C_ to transition from mono- to bi-allelic expression. This might tackle the plateau but still would not explain the time delay for bi-allelic expression. Again, I would encourage the authors to explore more complex models using better statistical tests: this would enable them to, at least, propose better accounting of their experimental data. Depending on what model improves on data fitting, further experimental validation may or may not be warranted (it might be beyond the scope of the present work).

We agree with reviewer #1 that at long timescales differentiation could potentially decrease rates of *Bcl11b* switching. In fact, we observed that cells can become locked into a mono-allelic expressing state as they progress through development (Figure 6). To model this scenario, we fit a variant of the parallel *trans-cis* model, where activation of the second allele occurs at a rate *k’_C_* that is lower than that of the first allele *k_C_*, as described by the reviewer. All other parameters in this model stayed constant.

However, this variant parallel model, with (*k_C_* = 0.011; *k’_C_* = 0.0057) provided only a marginally better fit to data compared to the parallel model (Author response image 1, blue line versus green line; and calculated chi‐squared values). There were minor differences in the shapes of the rises, with a slightly greater inflection in the mono-allelic expression time course. However, these differences did not generate a significantly lower chi-squared value. Thus, we conclude that the present data cannot significantly discriminate between these two models.

**Author response image 1. respfig1:** A slower rate of bi-allelic activation does not significantly improve model fit in the parallel *trans-cis* model. A) Diagram showing parallel *trans-cis* model shown in main text (left), and variant parallel *trans-cis* model with a slower rate of activation of the second *Bcl11b* allele (right). B) Best fits of cis-only (gray) sequential *trans-cis* (black), parallel *trans-cis* (green), and variant parallel *trans-cis* (blue) models to time evolution of the fraction of *Bcl11b* mono and bi-allelic expressing populations. See Figure 4 legend and text for description of fitting procedures. C) Bar chart showing the chi-squared value for the best fit for the four models, showing that variant parallel trans-cis model (forth column) does not significantly reduced chi-squared value compared to the simpler parallel *trans-cis* model (third column).

In summary, reviewer 1’s comments motivated an improved analysis that provides a better fit to the data and continues to show that the parallel *trans-cis* model is most consistent with the experimental data in Figure 4B. We fully agree with reviewer #1 that the underlying epigenetic processes are likely to be subject to more complex regulation, something we now note explicitly in our paper (subsection “A parallel *trans*-acting step enables expression from an activated *Bcl11b* locus”, first two paragraphs); however, technical limitations of the current experimental system – including fluorescent protein sensitivity and time delays in expression and accumulation – do not allow us to more accurately measure finer aspects of this regulation and draw any higher‐order conclusions.

Reviewer 2:[…] The findings are very interesting. The data indicate a very long activation time constant associated with all-or-none irreversible activities. I think this reflects the repositioning of the Bcl11b locus from the lamina to the nuclear interior. The authors kind of discuss this possibility but it could be described more precisely.

We agree that the repositioning of *Bcl11b* from the lamina to the nuclear interior, as convincingly shown in Isoda et al., 2017 could release the *Bcl11b* locus from the repressive environment of the nuclear lamina, and thus be a key part of the all‐or‐none activation mechanism. We have updated our Discussion to more explicitly describe this possibility:

“*Trans*‐regulators of DNA loop extrusion that associate with the distal enhancer, whose binding may be facilitated by non long-coding RNA transcription (Isoda et al., 2017), may stabilize these looping interactions (Fudenberg et al., 2016; Nasmyth, 2001; Riggs, 1990; Sanborn et al., 2015), which may release *Bcl11b* from the repressive environment of the nuclear periphery and permit its activation (Isoda et al., 2017).”

Along the same line the authors note "As Bcl11b turn on, its promoter establishes new contacts with the distal enhancer, resulting in de novo formation of an altered topological domain…". I think it is the other way around.

We agree completely with reviewer 2 that altered *Bcl11b* conformation likely precedes activation of its expression. Our previous statement meant to imply that *Bcl11b* expression correlates with altered conformation without implying causality; however, we see how this wording could have been misleading. We therefore changed the wording to avoid this confusion:

“As another possibility, the distal enhancer could recruit *trans*-factors that facilitate its T-lineage-specific looping with the *Bcl11b* promoter and its subsequent activation (Li et al., 2013). In early T‐cell progenitors, the *Bcl11b* promoter establishes new contacts with its distal enhancer, resulting in de novo formation of an altered topological associated domain, with boundaries defined by these two elements (Hu et al., 2018; Isoda et al., 2017).”

Reviewer 3:[…] • To my knowledge, the only other system to have been sufficiently studied to say anything in detail about the cis/trans logic is the FLC silencing system in Arabidopsis. I think it would be insightful to compare and contrast the two systems in more detail. FLC incorporates a cold-induced upstream trans upregulation, followed by two sequential cis steps (nucleation and spreading), where again the first cis step is very slow and stochastic. This Polycomb silencing system is in contrast to the parallel cis/trans activating switch at Bcl11b. Clearly there will be variety in the choice of cis/trans control and probably these two arrangements are just scratching the surface of the possible switching set-ups. But can the authors nevertheless comment more on the advantages of the particular trans/cis circuitry that appears to be implemented at Bcl11b?

We thank reviewer #3 for suggesting discussion and comparison of the Flc and *Bcl11b* systems, and wholeheartedly agree that the two systems provide an illuminating point of comparison and contrast. We have added a new paragraph in the Discussion on the relationship between the two systems (fourth paragraph).

With regard to FLC, I would also cite Yang et al., 2017, as that is the only other paper to my knowledge to implement two colour labelling with the goal of distinguishing cis/trans regulation.

Thank you. We now cite Yang et al. in the Introduction (second paragraph), as well as in the Discussion section (fourth paragraph).

• In the section where modelling is used to probe the function of Notch, I found the intuition for why perturbing the cis/trans steps changed the ratio of mono- versus bi-allelic expressing cells to be unclear. Could this be made clearer?

We agree that the intuition could have been clearer here. Briefly, because bi‐allelic cells need to undergo two successive *cis*-activation events, reducing the rate of *cis*‐activation would, in general, reduce the number of bi-allelic expressing cells relative to the number of mono-allelic expressing cells. Experimentally, Notch reduction has the opposite effect, increasing the number of bi-allelic cells relative to the number of mono-allelic cells, as expected for perturbations to the *trans* step. We now explain this in more detail through edits in the text as well as a new paragraph (subsection “Notch signaling controls the parallel *trans*-acting step in *Bcl11b* activation”, sixth paragraph).

Reviewing Editor:How well does the in vitro culture system that images individual DN2 cells as they divide and differentiate in the OP-9 system capture the developmental and gene dynamics of these cells in their thymic niche?In Figure 1 using their dual Bcl11b reporter system the authors show that DN2A progenitors in the mouse thymus are heterogeneous, approx. 30% manifest mono-allelic whereas approx. 40% evidence bi-allelic expression. DN2B cells, in contrast manifest predominantly bi-allelic expression (98%). There are two distinct interpretations of these initial observations. (i) DN2A cells undergo stochastic mono-allelic activation of Bcl11b and then go on to activate the second allele. (ii) the DN2A compartment is heterogeneous in its induction of Bcl11b with 40% of the cells simultaneously inducing both alleles and 30% of the cells exhibiting mono-allelic expression. The former cells are at a developmental advantage and outcompete the others in the DN2b compartment. If the latter interpretation is correct then it would suggest that in vivo the simultaneous bi-allelic activation path is favored. Furthermore the in vitro system being used to analyze the developmental activation of Bcl11b alleles favors the less efficient developmental path that DN2a cells can undertake. Another observation that seems to support this interpretation is that in vivo DN2a cells are suggested to convert into DN2b in three days and this is associated with consolidation of bi-allelic expression of Bcl11b. Figure 2D shows that even after 95 hr in culture approx. 60% of the DN2a cells have either failed to induce Bcl11b or manifest mono-allelic expression. Thus, it is possible that the in vitro system is biased toward the less efficient or minor developmental trajectory that the cells undertake in the thymus.

We agree with the reviewing editor that there are differences between *Bcl11b* activation kinetics in the thymus versus that on the OP9‐DL1 system, with a higher percentage of mono‐allelic cells seen at 4 days in OP9-DL1 culture compared to DN2B thymic progenitors freshly isolated out of the thymus. There are several things to say about this. (1) First, we regret that our original analysis in original Figure 1C did not correctly gate for DN2b cells, and therefore underestimated the percentage of mono‐allelic cells that persist in this population. (2) There are several reasons why we do not expect an exact concordance between Figure 1C and Figure 2D, based on known differences between the in vitro culture system and the thymic microenvironment. However, an overall speed difference would not change the argument that a *cis*‐regulatory constraint is needed to account for the differential expression of the two alleles in a single cell. (3) Also, while there is theoretically time for selection effects to operate in vitro, our experimental measurements show that if anything, the proliferation rate of cells that have fully turned on *Bcl11b* is slower, not faster, than those that have not yet turned it on. This makes it unlikely that a selective advantage for dual expressors could masquerade as differentiation. Detailed responses follow.

Unfortunately, our previous flow cytometry analysis in Figure 1C did not correctly gate for DN2b progenitors. Our initial gating captured cells that were too low for Kit expression to be DN2 cells, and would instead be more accurately classified as DN3 progenitors. As a result, our analysis substantially underestimated the fraction of mono‐allelic cells in the DN2b population.

We regret this error. We have re-done this analysis with correct gates for the DN2b population (Figure 1C). The flow plots of *Bcl11b*‐mCherry versus *Bcl11b*‐YFP levels from this population reveal a substantially higher fraction of *Bcl11b* mono-allelic cells (~15% instead of ~4%). Though these numbers are still lower than the percentage of mono-allelic *Bcl11b* expressing cells after 3 days of in vitro culture (Figure 2E, 40%), they indicate that the quantitative differences between allelic activation in thymus versus in vitro culture may not be as drastic as the initial data may suggest.

It has long been recognized that the OP9-DL1 culture system exaggerates the fraction of cells that can persist in DN2 (DN2a or DN2b) as compared to the fraction that remains in these stages in vivo (e.g. Huang et al.,2005, J Immunol). We are actively working to identify the transcription factors that may be regulated differently in vitro than in vivo (Scripture-Adams et al., 2014; W. Zhou, M. A. Yui, B. A. Williams, J. Yun, B. J. Wold, L. Cai, E. V. Rothenberg, unpublished data). In addition, it is also important to note that the snapshot of thymocyte subset phenotypes obtained in vivo is a steady‐state picture, not the representation of phenotypes of a single developmental cohort at a single timepoint, which we show in vitro. The average times that we cite for conversion from DN2a to DN2b, for example, come from in vitro data (Kueh et al., 2016), but in vivo the dwell time in DN2b may be considerably longer than in DN2a. Finally, when we allow cells to begin differentiation, then interrupt them for sorting to isolate a particular development subset before returning them to OP9-DL1 culture, we know there are stress effects that can slow progression speeds. Thus, there are numerous reasons why the absolute kinetics and the phenotypes of steady-state subset profiles in vivo may differ slightly from those seen at given timepoints in vitro, and that is why we include both kinds of data. However, differences in progression speeds between the two experimental systems do not change the need for a slow, *cis*‐acting mechanism to explain interallelic differences within single cells, which is seen in both systems.

We have included additional discussion of these discrepancies in the main text:

“The percentages of mono‐allelic cells generated at given timepoints on OP9‐DL1 co-culture differed from those in DN2b progenitors from the thymus, which have emerged from *Bcl11b* non-expressing DN2a cells at some unknown time in the past (40% versus ∼15%, Figure 1C). […] However, in both cases, *Bcl11b* mono-allelic as well as bi-allelic populations were clearly defined, indicating that the same slow *cis*‐activation processes observed in our experiments are also governing *Bcl11b* expression in the thymus.”

We are actively investigating the basis of this kinetic difference in ongoing work, for example in detailed comparisons of transcription factor dynamics in the Rothenberg lab (W. Zhou, M. A. Yui, J. Yun, L. Cai, E. V. Rothenberg, unpublished). Importantly, though, our existing data do not support the idea that bi‐allelic expressing *Bcl11b* progenitors have a greater rate of proliferation compared to mono‐allelic cells, as suggested by the Reviewing Editor. If anything, we have previously observed that *Bcl11b* activation coincides with a slow‐down of proliferation. in vivo, *Bcl11b*‐high cells move into G1 arrest at the next (DN3a) stage, enabling RAG‐mediated recombination at the TCR loci so as to qualify for β‐selection (Kueh et al., 2016; Yui et al., 2010). The *Bcl11b*‐high cells would thus be unlikely simply to outgrow the *Bcl11b*‐negative or *Bcl11b*‐low cells. Thus, it seems hard to escape the interpretation that *Bcl11b* mono-allelic cells differentiate into *Bcl11b* bi-allelic cells.

It is still possible that there are differences in the rates of *cis*‐activation, due to differences in the signaling environment between the thymus and the OP9‐DL1 system. Another possibility within the context of the parallel *trans-cis* model is that there is a large proportion of cells that have *cis*‐activated both *Bcl11b* alleles, but have not yet undergone the *trans*‐activation step. These cells would be expected to simultaneously turn on both alleles without passing through a mono-allelic intermediate. Elucidating the basis of these differences between development in the thymus and development on OP9-DL1 cultures will be an important direction for future experimental work.